



# Identification of linear response functions from arbitrary perturbation experiments in the presence of noise –

## Part I. Method development and toy model demonstration

Guilherme L. Torres Mendonça[1,2], Julia Pongratz[2,3], and Christian H. Reick[2]

[1]International Max Planck Research School on Earth System Modelling, Hamburg, Germany
[2]Max Planck Institute for Meteorology, Hamburg, Germany
[3]Ludwig-Maxmillians-Universität München, Munich, Germany

**Correspondence:** Guilherme L. Torres Mendonça (guilherme.mendonca@mpimet.mpg.de)

**Abstract.** Existent methods to identify linear response functions from data require tailored perturbation experiments, e.g. impulse or step experiments. And if the system is noisy, these experiments need to be repeated several times to obtain a good statistics. In contrast, for the method developed here, data from only a *single* perturbation experiment at *arbitrary* perturbation is sufficient if in addition data from an unperturbed (control) experiment is available. To identify the linear response function

for this ill-posed problem we invoke regularization theory. The main novelty of our method lies in the determination of the level of background noise needed for a proper estimation of the regularization parameter: This is achieved by comparing the frequency spectrum of the perturbation experiment with that of the additional control experiment. The resulting noise level estimate can be further improved for linear response functions known to be monotonic. The robustness of our method and its advantages are investigated by means of a toy model. We discuss in detail the dependence of the identified response function on

the quality of the data (signal-to-noise ratio) and on possible nonlinear contributions to the response. The method development presented here prepares in particular for the identification of carbon-cycle response functions in Part II of this study. But the core of our method, namely our new approach to obtain the noise level for a proper estimation of the regularization parameter, may find applications in solving also other types of linear ill-posed problems.

## 1 Introduction

To gain understanding of a physical system it is very helpful to know how it responds to perturbations. Considering a small time-dependent perturbation $f : \mathbb{R} \to \mathbb{R}$, the resulting time-dependent response $R : \mathbb{R} \to \mathbb{R}$ can from a very general point of view be written as

$$R(t) = \int_0^t \chi(t - s) f(s) ds, \qquad (1)$$

where the *linear response function* $\chi : \mathbb{R} \to \mathbb{R}$ is a characteristic of the considered system. In fact, under a number of assump-

tions – among which smoothness and causality are the most important –, Eq. (1) is the first term of a functional expansion of the





response $R$ into the perturbation $f$ around the unperturbed state $f(\cdot) = 0$, known as Volterra series (Volterra, 1959; Schetzen, 2010). In this framing, the key to gain insight into the system is the linear response function $\chi$: By knowing this function one has at hand not only a powerful tool to predict the response for sufficiently small but otherwise arbitrary perturbations, but also a means to study the internal dynamic modes of the unperturbed system by analyzing the temporal structure of the response

function.

Linear response functions arise within different contexts in many fields of science and technology. In physics, for example, material constants like the magnetic susceptibility or the dielectric function must be understood as linear response functions that by the Kubo theory of linear response (Kubo, 1957) can be expressed via the fluctuation-dissipation theorem by an auto-correlation of the unperturbed system. But applications of these functions range far beyond physics into fields like neurophysi-

ology and climate (Gottwald, 2020). The systems underlying these more modern applications are not covered by Kubo's theory valid only for near-equilibrium Hamiltonian systems. The most general result, covering also dissipative systems, stems from Ruelle (1997, 1998), who rigorously demonstrated the existence of linear response functions for uniform hyperbolic systems. But also non-hyperbolic systems may show a linear response (Reick, 2002; Lucarini, 2009) and it is believed that results for this class of systems transfer to large classes of nonequilibrium systems (Ruelle, 1999; Lucarini, 2008; Lucarini and Sarno, 2011;

Gallavotti, 2014; Ragone et al., 2016; Lucarini et al., 2017). In climate science, in particular, applications of linear response functions in the context of Ruelle's developments are a recent topic (e.g., Lucarini, 2009; Lucarini and Sarno, 2011; Lucarini et al., 2014; Ragone et al., 2016; Lucarini et al., 2017; Aengenheyster et al., 2018; Ghil and Lucarini, 2020; Lembo et al., 2020; Bódai et al., 2020). On the other hand, these functions have been successfully employed as a heuristic tool to study climate and the carbon cycle already for decades (e.g., Siegenthaler and Oeschger, 1978; Emanuel et al., 1981; Maier-Reimer and Hassel-

mann, 1987; Enting, 1990; Joos et al., 1996; Joos and Bruno, 1996; Thompson and Randerson, 1999; Pongratz et al., 2011; Caldeira and Myhrvold, 2013; Joos et al., 2013; Ricke and Caldeira, 2014; Gasser et al., 2017; Enting and Clisby, 2019). Yet another perspective is that from engineering sciences, in which the *impulse response* – that to a large extent corresponds to the linear response function – and the closely related *transfer function* (or *system function*) characterize linear time-invariant (LTI) systems, widely applied in fields such as signal processing and control theory (Kuo, 1966; Rugh, 1981; Beerends et al., 2003;

Boulet and Chartrand, 2006). Regardless of which viewpoint a particular community takes to investigate the linear response of a system, a fundamental step in this investigation is the identification of the appropriate linear response function.

In practical applications where the response function must be recovered from data, this may be a challenging task. The reason is that the identification problem is generally ill-posed so that by classical numerical methods one obtains a recovery severely deteriorated by noise (see below). In addition, existent methods to identify these functions from data require performing special

perturbation experiments. In the present study, we develop a method to identify linear response functions taking data from *any type* of perturbation experiment while fully accounting for the ill-posedness of the problem.

The generality of our method allows for deriving response functions in cases hardly possible before. Examples are problems where performing perturbation experiments is computationally expensive so that one must use data that were not designed for the purpose of deriving these functions. In geosciences, this may be the case when one is interested in characterizing by

response functions the dynamics of Earth System Models – extremely complex systems employed to simulate climate and





its coupling to the carbon cycle. In principle, with our method one can derive these functions taking simulation data from Earth-System-Models intercomparison exercises such as $C^4MIP$ – Coupled Climate-Carbon Cycle Model Intercomparison Project (Taylor et al., 2012; Eyring et al., 2016) – that are already available at international databases. In Part II of this study we explore this possibility by investigating in an Earth System Model the response of the land carbon cycle to atmospheric

$CO_2$ perturbations. Because of the relationship between the linear response function and the impulse response and the transfer function in LTI systems, our work can also be seen from the viewpoint of engineering sciences as a contribution to the corpus of methods to solve system identification problems (Åström and Eykhoff, 1971; Söderström and Stoica, 1989; Isermann and Münchhof, 2010; Pillonetto et al., 2014).

     In the field of climate science, the typical method to identify linear response functions is by means of the impulse response

function, which is the response to a Dirac delta-type perturbation (e.g., Siegenthaler and Oeschger, 1978; Maier-Reimer and Hasselmann, 1987; Joos et al., 1996; Thompson and Randerson, 1999; Joos et al., 2013). This method has become so widely known that often the terms linear response function and impulse response function are used interchangeably. Indeed, in the particular case where perturbations are weak, the two concepts coincide. But this is not true in general: If the impulse strength is large so that nonlinearities become important, the impulse response function differs from the linear response function.

Other studies have proposed to identify linear response functions by making use of other types of perturbations. Reick (2002) and Lucarini (2009) used a weak periodic forcing to derive response functions in the Fourier space (also called susceptibilities). Hasselmann et al. (1993), Ragone et al. (2016), MacMartin and Kravitz (2016), Lucarini et al. (2017), Van Zalinge et al. (2017), Aengenheyster et al. (2018) and Bódai et al. (2020) identify the linear response function using step experiments, where the perturbation is a Heaviside-type function.

As noted by Lucarini et al. (2014), in principle the linear response function of a system can be derived from an arbitrary type of perturbation experiment. One method would be to apply a Laplace transform to Eq. (1), so that $\chi(t)$ can in principle be computed by the inverse Laplace transform

$$\chi(t) = \mathcal{L}^{-1}\{\mathcal{L}\{R\}/\mathcal{L}\{f\}\}, \tag{2}$$

where $\mathcal{L}\{\cdot\}$ is the Laplace transform operator. In fact, a first step towards the derivation of $\chi(t)$ from the general Eq. (1) was

taken by Pongratz et al. (2011), although the problem was not systematically discussed.

     Deriving $\chi(t)$ from perturbation experiment data is not a trivial problem. For the general case where the perturbation is different from a Dirac delta-type function, the problem is ill-posed (e.g., Bertero, 1989; Landl et al., 1991; Lamm, 1996; Engl et al., 1996). This basically means that attempts to recover the exact $\chi(t)$ yield a solution with large errors due to an amplification of the noise in the data. On the other hand, when $f(t)$ is a Dirac delta-type function with sufficiently small

perturbation strength so that the response can be considered linear, the impulse response gives directly the linear response function, i.e. $\chi(t) = R(t)$. But even in this case noise may hinder the recovery: Because the perturbation is only one "pulse" with small perturbation strength, the response may have a too low signal-to-noise ratio because of internal variability (Joos et al., 2013), giving once more a recovery with large errors.





To remedy these noise problems, a method intended to "damp" the noise in the response is usually employed. In MacMartin
and Kravitz (2016), a step experiment with large perturbation strength is used to obtain a better signal-to-noise ratio in the
response, but at the cost of enhancing the effect of nonlinearities. An alternative approach is employed by Ragone et al. (2016)
and Lucarini et al. (2017), who employ an ensemble of simulation experiments and take the ensemble averaged response so
that the level of noise is reduced. But especially for complex models such as Earth System Models, ensembles of simulations
can be computationally extremely expensive so that such a procedure may not be feasible.

Instead of trying to improve the signal-to-noise ratio of the data by improved experiment design, here we are interested
in deriving $\chi(t)$ from a *single* realization of a *given* experiment by *accounting for the ill-posedness* of the problem. For this
purpose, we employ regularization theory. Although this theory offers a variety of methods to solve ill-posed problems (see
e.g.  Groetsch, 1984; Bertero, 1989; Bertero et al., 1995; Engl et al., 1996; Hansen, 2010), currently no general all-purpose
method exists. Typically, methods rely on some type of prior information about the problem (Istratov and Vyvenko, 1999).
Hence, they must be tailored according to the particularities of each application. Here, we develop a method requiring only
data from a given perturbation experiment and an unperturbed – or control – experiment (see section 3). The method solves the
ill-posed problem by applying Tikhonov-Phillips regularization. The regularization parameter is chosen via the discrepancy
method. An essential ingredient of the discrepancy method is the noise level, which is usually not known a priori. For this
reason, we propose a method to estimate the noise level by taking advantage of the information given by a spectral analysis of
the perturbation experiment *and* the control experiment. If the desired response function is known to be monotonic, the noise
estimate can be further adjusted. In section 4, the method is demonstrated to give reliable results under appropriate conditions
of noise and nonlinearity. In section 5, we compare the derived method with two existent methods in the literature to identify
the response function in the time domain. Results and technical details are discussed in section 6. Additional calculations are
shifted to appendices.

## 110 2  Linear response theory and basic ansatz of the method

As a preparation for introducing our method in section 3, in the present section we derive its basic ansatz, which takes into
account in addition to the response formula (1) also the noise in the data. Depending on the application context, the noise
may arise for different reasons, such as errors in the measurements, stochastic components in the system, etc. As will be seen,
our basic ansatz is in principle applicable to all those cases. But to make the connection to modern applications of linear
response functions that arise in the context of Ruelle's developments (e.g., climate), here we derive this ansatz starting from
considerations of linear response theory (Ruelle, 2009). Ruelle considered systems of type

$$\frac{d}{dt}\boldsymbol{x} = \boldsymbol{A}_0(\boldsymbol{x}) + \boldsymbol{A}_1(\boldsymbol{x})f(t), \tag{3}$$

where $\boldsymbol{x}(t)$ is the possibly infinite dimensional state vector and the perturbation $f(t)$ couples to the unperturbed system $\frac{d}{dt}\boldsymbol{x} = \boldsymbol{A}_0(\boldsymbol{x})$ via the field $\boldsymbol{A}_1(\boldsymbol{x})$. In the present context Eq. (3) could e.g. represent the dynamics of the Earth system perturbed by
anthropogenic emissions $f(t)$. Considering an observable $Y(\boldsymbol{x})$, Ruelle proved that the ensemble average of its deviation from





the unperturbed system $\langle \Delta Y \rangle$ can be expanded in the perturbation $f(t)$:

$$\langle \Delta Y(\boldsymbol{x}(t)) \rangle = \int_0^t \chi(t-s)f(s)ds + \mathcal{O}(f^2), \tag{4}$$

where the order symbol $\mathcal{O}(f^2)$ represents terms that vanish in the limit $f(\cdot) \to 0$ faster than the leading linear term. This expansion describes the response of a system that is *noisy* as a result of its chaotic evolution: Starting from different initial states

one obtains different values for $\Delta Y(\boldsymbol{x}(t))$. Compared to Eq. (1), in Eq. (4) the linear response function does not describe the response in observables directly but only in their ensemble average, i.e. in an average over the initial states of the unperturbed system. For the recovery of linear response functions from numerical experiments, this would mean that one had to perform many experiments starting from different initial states to obtain the appropriate ensemble averages. Using tailored perturbations experiments, it was demonstrated in several studies (e.g., Ragone et al., 2016; Lucarini et al., 2017; Bódai et al., 2020) that

linear response functions can indeed be obtained in this way, but at the expense of a large numerical burden from the need to perform many experiments. Instead, the aim here is to obtain the linear response functions from a *given* experiment and only from a *single* realization. Since we are dealing with a single realization, Eq. (4) becomes

$$\Delta Y(t) = \int_0^t \chi(t-s)f(s)ds + \eta(t) + \mathcal{O}(f^2), \tag{5}$$

where $\eta(t)$ is a noise term that must show up as a consequence of dropping the ensemble average. In addition, we assume

linearity in the perturbation. As a consequence, the present study is based on the ansatz

$$\Delta Y(t) = \int_0^t \chi(t-s)f(s)ds + \eta(t), \tag{6}$$

where now the response $\Delta Y(t)$ is divided into a deterministic term $\int_0^t \chi(t-s)f(s)ds$ and a noise term $\eta(t)$.

The linearity assumption is by purpose: In the present approach to derive the linear response function (see next section), hereafter called *RFI method* (Response Function Identification method), we first use Eq. (6) to obtain $\chi(t)$ and justify the

linearity assumption a posteriori by analyzing how robustly the response can be recovered for different perturbation strengths. Dropping the nonlinear terms has the advantage that one can use the corpus of linear methods to derive $\chi(t)$ from Eq. (6). Note that in practice, however small the perturbation may be, the nonlinear terms do not vanish. Therefore, the contribution of nonlinearities is in this way distributed between $\chi(t)$ and $\eta(t)$, which will be different from the previous $\chi(t)$ and $\eta(t)$ in Eq. (5). How strongly nonlinearities affect the numerical identification of $\chi(t)$ depends on the estimation of $\eta(t)$, which is a

crucial part of our RFI method and the main novelty introduced here to deal with the ill-posedness of the problem to identify $\chi(t)$.

In addition, although we derived Eq. (6) starting from considerations of linear response theory, it is clear that this ansatz can also be employed in any other context where it may be assumed that the response formula (1) applies and that the data is contaminated by additive noise.



## 3 Identification of linear response functions from arbitrary perturbation experiments

In this section we derive the RFI method. As mentioned above, the aim of this method is to obtain the linear response function using data from a single realization of a given perturbation experiment. For this purpose, the estimation of the noise term $\eta(t)$ is an essential step, which requires additionally data from an unperturbed (control) experiment.

Starting from the ansatz (6), the method is based on the idea that the noise term $\eta(t)$ can be estimated using information on the internal variability from the control experiment in combination with a spectral analysis of the perturbation experiment. The identification of the linear response function proceeds as follows: First, we assume that the linear response function decays multi-exponentially, i.e. we take a functional form for $\chi(t)$. Second, Eq. (6) is discretized for application to the discrete set of time series data, which results in a matrix equation. Then, assuming that the solution obeys the Picard condition (see below), we estimate the high-frequency components of the noise term $\eta(t)$ in Eq. (6) via a spectral analysis of the matrix equation applied to the data from the perturbation experiment. Next, assuming that the spectral distribution of noise is similar in the control and in the perturbation experiment, we estimate also the low-frequency components of $\eta(t)$. The final estimate of $\eta(t)$ is then used in a regularization procedure to find an approximate solution for $\chi(t)$. In case $\chi(t)$ is known to be monotonic, the approximated solution is further adjusted by checking for monotonicity.

In the first subsection, we introduce the assumption for the functional form of the linear response function. In subsections 3.2 and 3.3, we present the discretized problem and the solution using a regularization technique. In subsection 3.4 we derive the method to estimate the regularization parameter by estimating the noise in the data. Finally, in subsection 3.5 we show how the solution can be further improved in the presence of a monotonicity constraint.

### 3.1 Functional form of the linear response function

In general, the identification of linear response functions from data may be performed either pointwise (e.g., Ragone et al., 2016) or assuming a functional form (e.g., Maier-Reimer and Hasselmann, 1987). Both approaches usually lead to an ill-posed problem, and therefore to similar difficulties to find the solution (see more details in subsection 3.3). Although the RFI method may be applied in either case, here we assume that the response function consists of an overlay of exponential modes. By this ansatz we guarantee from the outset that the response relaxes to zero for $t \to \infty$, which is consistent with the expectation that real systems have finite memory. Besides constraining the function space for the derivation of the response function, another added value of this approach is that in principle it also gives the spectrum of internal time scales of the response.

Assuming this ansatz, the question on the functional form of $\chi(t)$ arises. In climate science, it is typically assumed that the response function can be described by only a few exponents (Maier-Reimer and Hasselmann, 1987; Enting and Mansbridge, 1987; Hasselmann et al., 1993, 1997; Grieser and Schönwiese, 2001; Li and Jarvis, 2009; Joos et al., 2013; Colbourn et al., 2015; Lord et al., 2016), i.e.

$$\chi(t) := \sum_{i=1}^{M} g_i e^{-t/\tau_i} \quad \text{with } M \text{ small,} \tag{7}$$





where the $\tau_i$ values are interpreted as characteristic time scales and the $g_i$ values are their respective weights. $\tau_i$ and $g_i$ are then obtained by applying some fitting technique taking a fixed number of terms $M$. Thus, an important step in this type of approach is to determine a suitable value for $M$. A common practice is to initially take only a small number of terms $M$, solve the problem and then add terms progressively, until the addition of a new term does not anymore improve the fit according

to some quality-of-fit criterion (e.g., Kumaresan et al., 1984; Maier-Reimer and Hasselmann, 1987; Hasselmann et al., 1993; Pongratz et al., 2011; Colbourn et al., 2015; Lord et al., 2016). Thereby it is assumed that once results stabilize the information in the data has been already fully exploited so that fitting of additional terms would be artificial. Nevertheless, finding the parameters $\tau_i$ and $g_i$ either from a given $\chi(t)$ by Eq. (7) or from $\Delta Y(t)$ by inserting Eq. (7) into Eq. (1) means to solve a special case of a Fredholm equation of the first kind (see Appendix A), which is an ill-posed problem (Groetsch, 1984). This

implies that even though the obtained solution may give a very good fit to the data, it may significantly differ from the exact solution (see e.g. the famous example from Lanczos, 1956, p. 272).

Therefore, to avoid the complication of determining $M$, we assume instead that the response function is characterized by a continuous spectrum $g(\tau)$ (Forney and Rothman, 2012):

$$\chi(t) = \int_0^\infty g(\tau) e^{-t/\tau} d\tau. \tag{8}$$

By this approach the time scale $\tau$ is not anymore an unknown, but is given after discretization by a prescribed distribution with $M$ terms covering a wide range of $\tau_i$ values. Thus, only a discrete approximation to the spectrum $g(\tau)$ needs to be found. In this way the functional representation is made independent from the question of information content as long as the spectrum of discrete time scales is chosen sufficiently large and dense to widely cover the spectrum of internal time scales of the considered system.

This approach has an additional advantage. By prescribing the distribution of time scales one must not solve a *nonlinear* ill-posed problem (by solving Eq. (7) for $\tau_i$ and $g_i$) but only a *linear* ill-posed problem (by solving Eq. (8) only for $g(\tau)$), for which the mathematical theory is fairly well developed (Groetsch, 1984; Engl et al., 1996). Because the problem is linear, the solution is even given analytically (see section 3.3), which makes the method very transparent.

### 3.2 Discretized problem

In view of applications to geophysical systems like the climate or the carbon cycle (Part II of this study) that are known to cover a wide range of time scales (Ghil and Lucarini, 2020; Ciais et al., 2013, Box 6.1), it is useful to switch to a logarithmic scale (Forney and Rothman, 2012) by rewriting Eq. (8) in terms of $\log_{10} \tau$:

$$\chi(t) = \int_{-\infty}^\infty q(\tau) e^{-t/\tau} d\log_{10} \tau, \quad \text{with} \quad q(\tau) := \tau \ln(10) g(\tau). \tag{9}$$

Hereafter, $q(\tau)$ and its discrete version $\boldsymbol{q}$ (see below) will be called *spectrum*.

In order to apply the basic Eq. (6) together with Eq. (9) to experiment data, the whole problem needs to be discretized in time and also with respect to the spectrum of time scales. Here we assume the data to be given at equally spaced time steps





$t_k = t_0 + k\Delta t$, $k = 0, 1, \ldots, N-1$, where $N$ is the number of data, while the time scales are assumed to be equally spaced at a logarithmic scale between maximum and minimum values $\tau_{max}$ and $\tau_{min}$, i.e.

$$\log_{10} \tau_j = \log_{10} \tau_{min} + j\Delta \log_{10} \tau, \quad j = 0, 1, \ldots, M-1,$$
$$\text{with} \quad \Delta \log_{10} \tau := \frac{\log_{10} \tau_{max} - \log_{10} \tau_{min}}{M} \tag{10}$$

where $M$ is the number of time scales. As shown in Appendix B, the resulting discretized equations corresponding to Eq. (6) and Eq. (9) are

$$\Delta Y_k = \Delta t \sum_{i=0}^{k} \chi_{k-i}\, f_i + \eta_k, \quad k = 0, \ldots, N-1, \tag{11}$$

and

$$\chi_k = \Delta \log_{10} \tau \sum_{j=0}^{M-1} q_j e^{-k\Delta t/\tau_j}, \quad k = 0, \ldots, N-1, \tag{12}$$

where $\eta_k$ stands for the noise. Combining the response data $\Delta Y_k$, the spectral values $q_j$, and the noise values $\eta_j$ into vectors $\boldsymbol{\Delta Y} \in \mathbb{R}^N$, $\boldsymbol{q} \in \mathbb{R}^M$, and $\boldsymbol{\eta} \in \mathbb{R}^N$, these equations can be written in vector form as

$$\boldsymbol{\Delta Y} = \mathbf{A}\boldsymbol{q} + \boldsymbol{\eta}, \tag{13}$$

with the components of matrix $\mathbf{A}$ given by

$$A_{kj} := \Delta \log \tau\, \Delta t \sum_{i=0}^{k} e^{-(k-i)\Delta t/\tau_j}\, f_i, \quad k = 0, \ldots, N-1, \; j = 0, \ldots, M-1. \tag{14}$$

The matrix $\mathbf{A}$ is known from the prescribed spectrum of time scales $\tau_i$ and the forcings $f_i$. Considering $\boldsymbol{\eta}$ as a fitting error, in principle one can apply standard linear methods to solve Eq. (13) for the desired spectrum by minimizing

$$\min_{\boldsymbol{q}_\eta} ||\mathbf{A}\boldsymbol{q}_\eta - \boldsymbol{\Delta Y}||^2, \tag{15}$$

where $||\cdot||$ denotes the Euclidean norm, i.e. $||\boldsymbol{x}|| = \sqrt{\sum_i x_i^2}$. Here we denoted the spectrum as $\boldsymbol{q}_\eta$ instead of $\boldsymbol{q}$ to emphasize that the spectrum found in this way can only be an approximation to the original $\boldsymbol{q}$ depending on the noise present in the data.

Unfortunately, it turns out that solving Eq. (15) is not a trivial task. The first difficulty is that the finite information provided by the data makes the problem underdetermined: Ideally one wants to obtain a spectrum $q(\tau)$ defined for $\tau \in [0, +\infty[$, but the data $\boldsymbol{\Delta Y}$ is discrete and covers only a limited time span. However, the most serious issue in identifying $\chi(t)$ arises because Eq. (1) is a special case of a Fredholm equation of the first kind (Groetsch, 1984, 2007, see also Appendix A), where the quest for the integral kernel is well-known to be an ill-posed problem (see e.g. Bertero, 1989; Hansen, 1992). This basically

means that any solution $\boldsymbol{q}_\eta$ of Eq. (15) obtained via classical numerical methods such as LU or Cholesky decomposition will be extremely sensitive to even small errors in the data (Hansen, 1992). Therefore, to solve Eq. (15) for the spectrum $\boldsymbol{q}_\eta$ we invoke regularization.





### 3.3 Regularized solution

As discussed above, the problem of solving Eq. (15) must be expected to be underdetermined and ill-posed. Strictly, the
underdetermination cannot be overcome. By contrast, the ill-posed nature of the problem can be treated in the application
investigated here. To deal with the ill-posedness, it is useful to perform a Singular Value Decomposition (SVD) of the matrix
$\mathbf{A}$:

$$\mathbf{A} = \mathbf{U}\boldsymbol{\Sigma}\mathbf{V}^T \tag{16}$$

with $\mathbf{A} \in \mathbb{R}^{N \times M}$, $\mathbf{U} \in \mathbb{R}^{N \times N}$, $\boldsymbol{\Sigma} \in \mathbb{R}^{N \times M}$, and $\mathbf{V} \in \mathbb{R}^{M \times M}$. $\boldsymbol{\Sigma}$ is a diagonal (not necessarily square) matrix with diagonal
entries $\sigma_0 \geq \sigma_1 \geq ... \geq \sigma_{M-1} \geq 0$ known as singular values, and

$$\mathbf{U} =: [\boldsymbol{u}_0, \boldsymbol{u}_1, ..., \boldsymbol{u}_{N-1}], \tag{17}$$
$$\mathbf{V} =: [\boldsymbol{v}_0, \boldsymbol{v}_1, ..., \boldsymbol{v}_{M-1}] \tag{18}$$

are orthonormal matrices with $\boldsymbol{u}_0, \boldsymbol{u}_1, ..., \boldsymbol{u}_{N-1}$ being the left singular vectors and $\boldsymbol{v}_0, \boldsymbol{v}_1, ..., \boldsymbol{v}_{M-1}$ the right singular vectors
of $\mathbf{A}$. In practice, assuming that there is more data than prescribed time scales, i.e. $N \geq M$ the singular values $\sigma_i$ computed
numerically are nonzero (see Golub and Van Loan, 1996, section 5.5.8). In this case, Eq. (15) has the unique solution (see
Golub and Van Loan, 1996, Theorem 5.5.1)

$$\boldsymbol{q}_\eta = \sum_{i=0}^{M-1} \frac{\boldsymbol{u}_i \bullet \boldsymbol{\Delta Y}}{\sigma_i} \boldsymbol{v}_i, \tag{19}$$

where $\bullet$ denotes the usual scalar product.

In practice, when a SVD is applied to a discrete version of a Fredholm equation of the first kind, the components of the
singular vectors $\boldsymbol{v}_i$ and $\boldsymbol{u}_i$ tend to have more sign changes with increasing index $i$, as observed by Hansen (1989, 1990). This
observation justifies that in the following we dub low-index terms in Eq. (19) as *low-frequency* contributions, and high-index
terms as *high-frequency* contributions.

It is well-known that when applying solution (19) one encounters certain numerical problems. Regularization is a means to
handle these problems. These problems arise – even in the absence of noise – as follows. From the Riemann-Lebesgue lemma
(see e.g. Groetsch, 1984) it is known that the high-frequency components of the data $\Delta Y(t)$ must approach zero. In the discrete
case, by Hansen's observation this means that the projections $\boldsymbol{u}_i \bullet \boldsymbol{\Delta Y}$ should approach zero for increasing index values $i$. But
due to machine precision or the noise $\boldsymbol{\eta}$ contained in $\boldsymbol{\Delta Y}$, numerically the absolute values $|\boldsymbol{u}_i \bullet \boldsymbol{\Delta Y}|$ do not approach zero
but settle at a certain non-zero level for large $i$ or, in the presence of noise, may even increase. Due to the ill-posedness also
the singular values $\sigma_i$ in the denominator of Eq. (19) tend to zero so that these high-frequency contributions to $\boldsymbol{q}_\eta$ are strongly
amplified. Hence applying Eq. (19) naively would not give a stable solution for $\boldsymbol{q}_\eta$ because its value would depend critically
on numerical errors and the noise present in the data.

Regularization remedies this problem by suppressing the problematic high-frequency components. This approach assumes
that the main information on the solution is contained in the low-frequency components so that the high-frequency contributions





to the sum (19) can be ignored. This assumption is consistent with the very nature of ill-posed problems because in such
problems information on high frequencies is anyway supressed so that only low-frequency components of the solution are
recoverable (Groetsch, 1984, section 1.1).

To perform such filtering, we employ the Tikhonov-Phillips Regularization method (Phillips, 1962; Tikhonov, 1963) –
although the method is most famously known as Tikhonov regularization, here we choose this different name in recognition
of Phillips earlier work (see Groetsch, 2003). Besides being mathematically well-developed (see e.g. Groetsch, 1984; Engl
et al., 1996), the Tikhonov-Phillips Regularization method gives an explicit solution in terms of the SVD expansion, which
allows for a clear interpretation of the filtering. In addition, it provides a smooth filtering of the solution, in contrast to the
also well-known Truncated Singular Value Decomposition method (Hansen, 1987). For additional regularization methods, see
e.g. Bertero (1989), Bertero et al. (1995) and Palm (2010).

The standard Tikhonov-Phillips Regularization yields the regularized solution in the simple form (Hansen, 2010; Bertero,
1989)

$$\boldsymbol{q}_\lambda = \sum_{i=0}^{M-1} f_i(\lambda) \frac{\boldsymbol{u}_i \bullet \boldsymbol{\Delta Y}}{\sigma_i} \boldsymbol{v}_i, \qquad (20)$$

where the $f_i(\lambda)$ are the filter functions

$$f_i(\lambda) = \frac{\sigma_i^2}{\sigma_i^2 + \lambda}. \qquad (21)$$

By adding the filter function indeed the high-frequency components are suppressed: With $\lambda$ properly chosen, at large index $i$,
where $\sigma_i^2 \ll \lambda$, $f_i(\lambda)$ approaches $\sigma_i^2/\lambda$ so that the terms under sum sign are proportional to $\sigma_i$ meaning that the terms for large
$i$ do not contribute significantly to the sum. In contrast, for small $i$, $\lambda \ll \sigma_i^2$ so that $f_i(\lambda)$ is about 1 and the terms under the
sum are almost unchanged. In this way the filter function indeed selects only the low-frequency components. Therefore, now
the problem boils down to determining $\lambda$ (see next section). Once $\lambda$ is determined, the solution $\boldsymbol{q}_\lambda$ is obtained by Eq. (20) and
the desired linear response function $\chi(t)$ finally follows from Eq. (12).

### 3.4 Determining the regularization parameter $\lambda$ from the noise

By construction it is clear that $\boldsymbol{q}_\lambda$ as computed from Eq. (20) strongly depends on the regularization parameter $\lambda$. Accordingly,
much effort has been put in developing methods to determine suitable values for $\lambda$ (e.g., Engl et al., 1996; Hansen, 2010). Of
special interest are methods that give solutions converging with decreasing noise level to the "true" solution. One such method
known to conform to this condition while uniquely determining the regularization parameter has been proposed by Morozov
(1966). His *discrepancy method* is based on the idea that the solution to the problem allows the data to be recovered with an
error of the magnitude of the noise (Groetsch, 1984): Let $\delta$ denote an upper bound of the *noise level* $||\boldsymbol{\eta}||$, i.e. $\delta \geq ||\boldsymbol{\eta}||$. Then,
$\lambda$ should be chosen such that the discrepancy matches $\delta$, i.e.

$$||\mathbf{A}\boldsymbol{q}_\lambda - \boldsymbol{\Delta Y}|| = \delta. \qquad (22)$$




Groetsch (1983) motivates the choice of this method by demonstrating that determining $\lambda$ from Eq. (22) minimizes a natural

choice for an upper bound of the error in the solution given by regularization. Unfortunately, the noise level $\delta$ is usually not known. But in the following we assume that data from an unforced experiment (control experiment) are available – as is typically the case in applications to Earth System Models (see Part II) – that allow for an independent estimate of the noise level $\delta$.

A naive way to invoke these data to determine $\lambda$ would be to take $\delta$ essentially as the standard deviation of the control

experiment – more precisely: $\delta := \sigma\sqrt{N} = ||\boldsymbol{\Delta Y}_{ctrl}||$. Technically, to find $\lambda$, one may start with a large value for $\lambda$ and decrease it until the left hand side of Eq. (22) matches $\delta$ (as suggested by Hämarik et al., 2011). That this procedure works is explained by the fact that the function $\lambda \mapsto ||\mathbf{A}\boldsymbol{q}_\lambda - \boldsymbol{\Delta Y}||$ is continuous, increasing and contains $\delta$ in its range (Groetsch, 1984, Theorem 3.3.1). Having found $\lambda$ in this way, the desired solution $\boldsymbol{q}_\lambda$ is then obtained from Eq. (20). But this approach is not as straightforward as one may think: Because of the forcing, the noise in the perturbed experiment may have different

characteristics from that in the control experiment. Therefore in the following we devise a method how to account for this problem.

Formally in Eq. (13) $\boldsymbol{\Delta Y}$ is split into a "clean" part and noise $\boldsymbol{\eta}$. Entering this into Eq. (19) gives

$$\boldsymbol{q}_\eta = \sum_{i=0}^{M-1} \left( \frac{\boldsymbol{u}_i \bullet \mathbf{A}\boldsymbol{q}}{\sigma_i} \boldsymbol{v}_i + \frac{\boldsymbol{u}_i \bullet \boldsymbol{\eta}}{\sigma_i} \boldsymbol{v}_i \right). \tag{23}$$

Accordingly, the first term in the sum gives the "true" solution $\boldsymbol{q}$ while the second term gives the noise contribution to $\boldsymbol{q}_\eta$.

As already pointed out when discussing regularization, the "true" solution of ill-posed problems can only be recovered if it is dominated by the projection onto the first singular vectors. This requirement is formally stated by the discrete Picard condition (Hansen, 1990), which demands that the size of the projection coefficients $|\boldsymbol{u}_i \bullet \mathbf{A}\boldsymbol{q}|$ drops sufficiently fast to zero so that they get smaller than $\sigma_i$ before $\sigma_i$ levels off to a finite value because of numerical errors. To find a good estimate for the noise level $\delta$ we use this in the following way. Let $i_{max}$ be the value of the index $i$ where the singular values $\sigma_i$ start to level off. Assuming

that the Picard condition holds, one can infer that

$$\frac{\boldsymbol{u}_i \bullet \boldsymbol{\Delta Y}}{\sigma_i} \overset{(13)}{=} \frac{\boldsymbol{u}_i \bullet \mathbf{A}\boldsymbol{q}}{\sigma_i} + \frac{\boldsymbol{u}_i \bullet \boldsymbol{\eta}}{\sigma_i} \approx \frac{\boldsymbol{u}_i \bullet \boldsymbol{\eta}}{\sigma_i} \quad \text{for } i > i_{max}. \tag{24}$$

This conclusion follows because for $i > i_{max}$ the first term after the equal sign has already dropped towards zero because of the Picard condition, while the projections of the noise $\boldsymbol{u}_i \bullet \boldsymbol{\eta}$ in the next term are finite, meaning that the first term is vanishing in relation to the second term. Therefore

$$\boldsymbol{u}_i \bullet \boldsymbol{\Delta Y} \approx \boldsymbol{u}_i \bullet \boldsymbol{\eta} \quad \text{for } i > i_{max}. \tag{25}$$

This equation determines the high-frequency components of the noise $\boldsymbol{\eta}$. It remains to determine also the low-frequency components to obtain an estimate for $\delta$.

For this purpose, we take advantage of the data from the control experiment. The control experiment is an experiment performed for the same conditions as the perturbed experiment, with the only difference that the forcing $\boldsymbol{f}$ is zero so that the

resulting $\boldsymbol{\Delta Y}_{ctrl}$ can be understood as pure noise; therefore we write $\boldsymbol{\eta}_{ctrl} := \boldsymbol{\Delta Y}_{ctrl}$. While in the forced experiment the





low-frequency noise is obscured by the low-frequency response induced by the forcing, the low-frequency part of the control experiment data can to first order be expected to give an estimate of the low-frequency noise present in the forced experiment. Nevertheless, it is clear that due to the forcing the spectral characteristics of noise may be different in the forced and unforced experiments. More precisely, the spectrum of noise may differ in its *overall level* and *spectral distribution* (i.e. the "shape" of

the spectrum). In the following, we account for a possible difference in the overall level. However, we will assume that the spectral distribution is approximately the same for $\boldsymbol{\eta}_{ctrl}$ and $\boldsymbol{\eta}$; we call this the *spectral similarity assumption*.

After these considerations, $\lambda$ can be determined as follows: Take $i_{max}$ as the last index $i$ before the plateau $\sigma_i \approx 0$. This $i_{max}$ distinguishes high-frequency ($i > i_{max}$) from low-frequency ($i \leq i_{max}$) components. Then

$$z := || \left[ \boldsymbol{u}_{i_{max}+1} \bullet \boldsymbol{\Delta Y}, ..., \boldsymbol{u}_{M-1} \bullet \boldsymbol{\Delta Y} \right]^T ||, \tag{26}$$

$$z_{ctrl} := || \left[ \boldsymbol{u}_{i_{max}+1} \bullet \boldsymbol{\eta}_{ctrl}, ..., \boldsymbol{u}_{M-1} \bullet \boldsymbol{\eta}_{ctrl} \right]^T || \tag{27}$$

are the levels of high-frequency noise in the perturbed (see Eq. (25)) and in the control experiment, respectively. We now scale the spectral components of $\boldsymbol{\eta}_{ctrl}$ so that its high-frequency level matches the high-frequency level of $\boldsymbol{\Delta Y}$:

$$\boldsymbol{\eta}' := \frac{z}{z_{ctrl}} \boldsymbol{\eta}_{ctrl}. \tag{28}$$

In this way, the magnitude of the high-frequency components of $\boldsymbol{\eta}'$ matches that of $\boldsymbol{\Delta Y}$, and because of Eq. (25) also that

of $\boldsymbol{\eta}$. On the other hand, the spectral distribution of $\boldsymbol{\eta}'$ is the same as for $\boldsymbol{\eta}_{ctrl}$, and by the spectral similarity assumption approximately the same as for $\boldsymbol{\eta}$. Because $\boldsymbol{\eta}'$ and $\boldsymbol{\eta}$ have similar spectral distributions, the fact that the magnitude of the high-frequency components of $\boldsymbol{\eta}'$ matches that of $\boldsymbol{\eta}$ implies that also the magnitude of their low-frequency components matches. Therefore, $\boldsymbol{\eta}'$ can be seen as an estimate of the noise $\boldsymbol{\eta}$ in the perturbed system not only at high but also at low frequencies. Hence this corrected noise vector $\boldsymbol{\eta}'$ can be used to obtain an estimate of the noise level of the perturbed system by setting

$$\delta := ||\boldsymbol{\eta}'||. \tag{29}$$

Compared to taking for $\delta$ simply the noise level from the unperturbed experiment (as was insinuated above), taking it in this scaled way assures that the high-frequency components are consistent with the Picard condition that must hold for $\boldsymbol{q}$ to be recoverable from the ill-posed problem tackled here. Having determined $\delta$, $\lambda$ can now be computed from Eq. (22) as described above, from which the $\boldsymbol{q}$ follows (Eq. (20)) and hence $\chi(t)$ (Eq. (12)).

### 3.5 Additional noise level adjustment in the presence of a monotonicity constraint

In the application to the land carbon cycle in Part II of this study we show that certain response functions $\chi(t)$ decrease monotonically to zero. In attempts to recover such response functions by employing the noise level adjustment described in the previous section, it may turn out that the numerically found response function fails to be monotonic. There may be several reasons for this failure (strong nonlinearities, signal too obscured by noise, etc.). But one additional reason may be that the low-

frequency level of the noise was not properly estimated by assuming that the spectral distribution in the unperturbed experiment





reflects the distribution in the perturbed experiment. For such cases one may try to improve the result by further adjustment of the low-frequency noise level to obtain a more reasonable result.

The idea is to adjust the low-frequency components of noise independently of the high-frequency components iteratively until the solution obeys the monotonicity constraint. To understand how to do so, several things must be explained:

1. A sufficient condition for $\chi(t)$ being monotonic is that all components $q_i$ have the same sign (see Appendix C). Therefore, starting out from a numerical solution for $\chi(t)$, it would develop towards monotonicity if one could come up with a sequence of vectors $\boldsymbol{q}_\lambda$ having less and less sign changes.

2. From Eq. (20) it is seen that because of Hansen's observation explained in section 3.3, that singular vectors $\boldsymbol{v}_i$ are less noisy for lower $i$, $\boldsymbol{q}_\lambda$ has fewer sign changes the fewer $\boldsymbol{v}_i$ contribute to the sum.

3. As seen from Eq. (20) and Eq. (21), this is the case the more components the filter function is suppressing, i.e. the larger the value of $\lambda$.

4. To obtain larger values of $\lambda$, one sees from the discrepancy method (22) that one has to increase $\delta$. The proof for this can be found in Groetsch (1984) (Theorem 3.3.1) but it can also be made plausible as follows: Starting from $\lambda = 0$, $\boldsymbol{q}_\lambda = \boldsymbol{q}_\eta$, which is the solution of the minimization problem (15). Hence, for $\lambda = 0$ the discrepancy in the left-hand side of Eq. (22) is minimal.

By increasing $\lambda$ one decreases all components of $\boldsymbol{q}_\lambda$ (Eq. (20)), increasing thereby the discrepancy.

5. Following the reasoning of the previous section, in order to obtain a larger value for $\delta$ one must increase the noise level $||\boldsymbol{\eta}'||$ (compare Eq. (29)). In doing so, one must keep the high-frequency components of $||\boldsymbol{\eta}'||$ unchanged because they must keep matching the level of the high-frequency components of the noise in the perturbed experiment $\boldsymbol{\eta}$ (given by Eq. (25)). Hence, to increase $\delta$ one sees from Eq. (29) that this is achieved by scaling up the low-frequency components of $||\boldsymbol{\eta}'||$.

Summarizing these considerations, we have to increase the level of low-frequency contributions to $\boldsymbol{\eta}'$ to develop a given solution for $\chi(t)$ towards monotonicity.

This leads to the overall algorithm listed in Fig. 1. The first five steps have been already explained at the end of section 3.4. To account for monotonicity the additional step 6 combined with the loop back to step 4 has to be iteratively executed. To enhance the low-frequency noise level as explained above, we calculate in step 6 a new noise vector $\boldsymbol{\eta}_{new}$ by keeping the

high-frequency part from $\boldsymbol{\eta}$ and enhancing its low-frequency components by a factor $c > 1$. Then we recompute $\chi(t)$ from steps 4 and 5 and once more check for monotonicity.

# 4   Applicability in the presence of noise and nonlinearities

In application to real data the presence of noise and nonlinearities may complicate the recovery of linear response functions. Therefore, by using artificial data generated from a toy model, in the present section we analyze the robustness of the RFI

method in the presence of such complications. Robustness for real data is studied in Part II.



1. Take $\boldsymbol{\eta}_{ctrl} := \boldsymbol{\Delta Y}_{ctrl}$ from the control experiment.

2. Determine $i_{max}$ as the last $i$ before the plateau $\sigma_i \approx 0$.

3. Define

$$z := || \left[ \boldsymbol{u}_{i_{max}+1} \bullet \boldsymbol{\Delta Y}, ..., \boldsymbol{u}_{M-1} \bullet \boldsymbol{\Delta Y} \right]^T ||,$$
$$z_{ctrl} := || \left[ \boldsymbol{u}_{i_{max}+1} \bullet \boldsymbol{\eta}_{ctrl}, ..., \boldsymbol{u}_{M-1} \bullet \boldsymbol{\eta}_{ctrl} \right]^T ||.$$

Set

$$\boldsymbol{\eta}' := \frac{z}{z_{ctrl}} \, \boldsymbol{\eta}_{ctrl} \qquad \text{(spectral similarity assumption)}.$$

4. Set $\quad \delta := ||\boldsymbol{\eta}'||, \quad$ solve Eq. (22) for $\lambda$, and obtain $\boldsymbol{q}_\lambda$ from Eq. (20).

5. Compute $\chi(t)$ from Eq. (12).

This is the final result except a monotonicity should be accounted for. In that case the algorithm proceeds as follows:

6. Check if the resulting $\chi(t)$ decays monotonically to zero. If so, we are done. Else, enhance the low-frequency noise level by setting

$$\boldsymbol{\eta}_{new} := c \sum_{i=0}^{i_{max}} \boldsymbol{u}_i \bullet \boldsymbol{\eta}' \boldsymbol{u}_i + \sum_{i=i_{max}+1}^{M-1} \boldsymbol{u}_i \bullet \boldsymbol{\eta}' \boldsymbol{u}_i,$$

where $c$ is some value larger than 1. Then set $\boldsymbol{\eta}' := \boldsymbol{\eta}_{new}$ and repeat calculations starting from step 4.

**Figure 1.** Final RFI algorithm (see text for details).

### 4.1 Toy model and artificial experiments

As toy model we take

$$\frac{d}{dt}\boldsymbol{x}(t) = M\boldsymbol{x}(t) + f(t)\boldsymbol{a} + \boldsymbol{n}(t). \tag{30}$$

Here the matrix $M \in \mathbb{R}^{D \times D}$ describes the relaxation of the unperturbed model. The second right-hand side term represents the deterministic forcing constructed from the time-dependent forcing strength $f \colon \mathbb{R} \to \mathbb{R}$ and the coupling vector $\boldsymbol{a} \in \mathbb{R}^D$. Additionally, the system is perturbed by the stochastic forcing $\boldsymbol{n} \colon \mathbb{R} \to \mathbb{R}^D$, which for simplicity is assumed to be white noise. To make the relation to the carbon cycle considered in Part II, the components of $\boldsymbol{x}$ may be understood as the carbon stored in plant tissues and soils at the different locations worldwide, so that the observable $Y(t) := \sum_i x_i(t)$ is the analogue of globally





stored land carbon. The solution of the system is

$$
\boldsymbol{x}(t) = \int_0^t e^{M(t-s)} \boldsymbol{a} f(s) ds + \int_0^t e^{M(t-s)} \boldsymbol{n}(s) ds. \tag{31}
$$

Since almost every linear system can be diagonalized, we assume from the outset $M$ to be diagonal with eigenvalues $-1/\tau_i^*$, the $\tau_i^*$ being the relaxation time scales. Then

$$
Y(t) = \int_0^t \chi^*(t-s) f(s) ds + \eta^*(t) \tag{32}
$$

with the linear response function $\chi^*(t)$ and the noise term $\eta^*(t)$ given by

$$
\chi^*(t) = \sum_{i=0}^{D-1} a_i e^{-t/\tau_i^*}, \tag{33}
$$

$$
\eta^*(t) = \sum_{i=0}^{D-1} \int_0^t e^{-(t-s)/\tau_i^*} n_i(s) ds. \tag{34}
$$

To complete the description of the toy model one has to specify its parameters. For the dimension $D$ we take 70 and the time scales are assumed to be distributed logarithmically between $\tau_{min}^* = 0.01$ and $\tau_{max}^* = 1000$, i.e. $\tau_i^* = 0.01 \times 10^{i \Delta \log \tau}$ with $\Delta \log \tau = (\log_{10} 10^3 - \log_{10} 10^{-2})/70$. With carbon cycle applications in mind, the distribution of the components of the coupling vector is adapted from the log-normal rate distribution found by Forney and Rothman (2012) for the decomposition of soils:

$$
a_i = \frac{1}{\tau_i^* \sigma \sqrt{2\pi}} \exp\left(-\frac{(\ln \tau_i^* - \mu)^2}{2\sigma^2}\right), \tag{35}
$$

with $\mu$ and $\sigma$ chosen so that the peak time scale is around $\tau = 5$ and the limits of the log-normal distribution are approximately within $\tau = 0.1$ and $\tau = 200$ (see "true" spectrum in Fig. 3). The components of $\boldsymbol{n}$ are are taken as uncorrelated, i.e. $\langle n_i(0) n_j(t) \rangle = \xi \delta_{ij} \delta(t)$, with standard deviation $\xi$ being chosen differently in different experiments.

In our experiments we explore how $Y(t)$ behaves as a function of the forcing $f(t)$. To this end, we choose a forcing function $f(t)$ (see Table 1 and Fig. 2). The most obvious way to perform the toy model experiments would be to integrate Eq. (30). But to have a better control over the noise it is for our purpose more appropriate to use the analytical solution (32)–(34). Hence, we numerically integrate Eq. (32), using the representation (33) and (34). The data from these experiments are then used to investigate the performance of the RFI algorithm to recover $\chi^*(t)$. Since all $a_i$ values are non-negative, the response function (33) is monotonic, so that we apply the extended version of the algorithm (see Fig. 1 including step 6). In all experiments we generate $N = 140$ data points, to have a time series of similar length as in the climate change simulations analyzed in Part II (140 years, one value for each year). To apply the RFI method also the noise from an associated control experiment is needed. This is obtained from Eq. (34) by using another realization $n_i(t)$ of white noise for each system dimension $i$.


**Table 1.** Experiments considered in this study. Forcings are shown in Fig. 2. To standardize the type of experiments considered here and in Part II, we select forcing functions that mimic those employed in climate change simulation experiments to whose data the RFI method is applied in Part II. Note that in principle any type of forcing could be employed.

| Type | Forcing | Description |
|---|---|---|
| Percent | 0.5% | |
| | 0.75% | |
| | 1% | Forcing is increased from a starting value at the specified |
| | 1.5% | percent rate per time step. |
| | 2% | |
| Step | $1.1 \times f_0$ | Forcing is abruptly increased from a starting value by the |
| | $2 \times f_0$ | specified factor. |
| Control | zero | Forcing is held fixed at zero. |

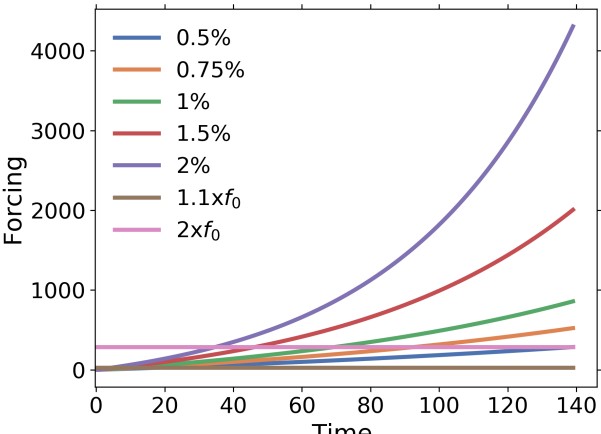

**Figure 2.** Forcings for the experiments considered in this study. To standardize the type of experiments considered here and in Part II, we select forcing functions that mimic those employed in climate change simulation experiments to whose data the RFI method is applied in Part II. Note that in principle any type of forcing could be employed.

## 4.2 Choice of parameters for the RFI method

To apply the RFI method, we choose $M = 30$ time scales for the recovery of $\chi^*$. Using $\tau_{min} = 0.1$ and $\tau_{max} = 10^5$ we distribute the spectrum of time scales according to Eq. (10). These parameters are also used for the application on the carbon cycle in Part II and for the comparison with previous methods in section 5.




### 4.3 Ideal conditions

To gain trust in the numerics of our implementation of the RFI method we present in this section a technical test considering conditions under which it is known that the linear response function should be quite perfectly recoverable. Such ideal conditions are characterized by perfect linearity and absence of noise. Hence we use the presented toy model (which is anyway linear) in the absence of noise ($\boldsymbol{n} = \boldsymbol{0}$) for this test. Actually, this will not be a full test of the algorithm, but only of the implementation of its basic apparatus (sections 3.2 and 3.3) culminating in Eq. (20) since in the absence of noise the method to determine the regularization parameter $\lambda$ (sections 3.4 and 3.5) is not applicable. One might think that in the absence of noise one could use Eq. (19) to determine the linear response function, but even under such ideal conditions the ill-posedness of the problem calls for regularization to suppress the *numerical noise* that prevents from obtaining a sensible solution from Eq. (19) (see discussion in the paragraph after Eq. (19)). But choosing the extremely small value of $\lambda = 10^{-8}$ for the regularization parameter when evaluating Eq. (20) is sufficient for this technical test.

Figure 3(c) shows the response of the noiseless toy model to the forcings shown in Fig. 2, i.e. we performed the experiments listed in Table 1, although for the present test the control experiment is not needed.

Applying Eq. (20) to the experiment data gives the spectrum $\boldsymbol{q}_\lambda$ shown in Fig. 3(a). Here, we derived the spectrum $\boldsymbol{q}_\lambda$ for each experiment separately, although in the figure only single dots are seen, because all results coincide so closely and are almost indistinguishable from the "true" solution $\boldsymbol{q}^*$ as was expected for this ideal case. The next Fig. 3(b) shows the response function obtained from the spectra $\boldsymbol{q}_\lambda$ using Eq. (12). Obviously from Fig. 3(a) the "true" response function is reconstructed perfectly from whatever experiment used. As a final test we predict using in Eq. (1) the response function obtained from the 1% experiment the responses of other experiments. And indeed, these predicted responses are indistinguishable from the responses obtained directly from the experiments (see Fig. 3(c)). This latter result demonstrates perfect robustness of the numerical approach to recover the responses in this ideal case.

### 4.4 First complication: noise

The presence of noise may severely hinder the detailed recovery of $\chi_*(t)$ due to the ill-posed nature of the problem. As already discussed in 3.3, small singular values of the matrix $\mathbf{A}$ tend to amplify the noise terms associated to the high-frequency singular vectors in the expansion (23). Tikhonov-Phillips regularization filters out these terms via Eq. (20) and Eq. (21). As a consequence, the accuracy of the solution $\boldsymbol{q}_\lambda$ tends to decrease as the noise level increases. How severely the accuracy of $\boldsymbol{q}_\lambda$ decreases depends on the smoothness of the "true" spectrum $\boldsymbol{q}^*$: The fewer terms one needs to describe $\boldsymbol{q}^*$ (i.e. the smoother it is), the higher is the level of noise at which it is still possible to recover $\boldsymbol{q}^*$ with acceptable accuracy. This will get more clear from the Picard plots introduced below when illustrating the effect of noise using experiments with the toy model.

In order to demonstrate the effect of the addition of noise on the quality of the derived $\chi(t)$, we define a relative error for the prediction of the responses from different experiments. Consider a particular experiment – which is in our case the 1% experiment – from which we have obtained by the RFI method the response function, which we call here $\chi^0(t)$. The relative



**Figure 3.** Demonstration of robust recovery for noise-free data from the toy model: (a) recovered $\boldsymbol{q}_\lambda$; (b) recovered $\chi(t)$; and (c) original responses and predictions using $\chi(t)$ derived from the 1% experiment. Reconstructed values are almost indistinguishable from original data. For plotting the "true" spectrum of the toy model in subfigure (a) we used the relation $\boldsymbol{q}^* = \boldsymbol{a}/\Delta \log_{10} \tau$, which can be obtained by comparing Eq. (33) with Eq. (12). Since from the discrete spectrum the response function and the response may be obtained for any time $t$, the spectrum is plotted as dots while the response function and response are plotted as continuous lines. The regularization parameter is chosen as $\lambda = 10^{-8}$.

error for the prediction of the response from an experiment "$k$" by the recovered $\chi^0(t)$ via the convolution (1) is

$$\varepsilon_k^0 := \frac{||\boldsymbol{\Delta Y}^k - \boldsymbol{\chi}^0 \star \boldsymbol{f}^k||}{||\boldsymbol{\Delta Y}^k||}, \tag{36}$$

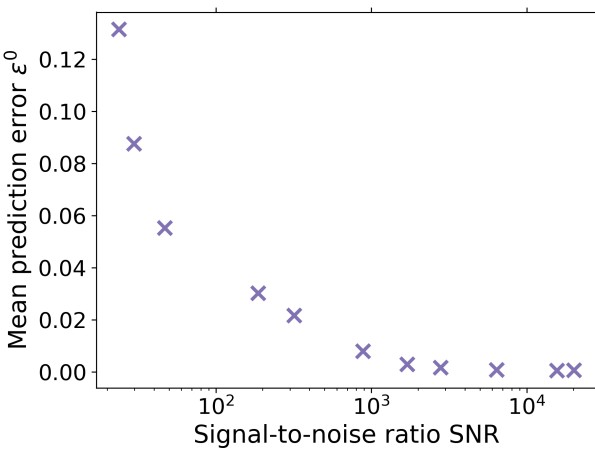

**Figure 4.** Mean prediction error (37) of the recovery when deriving $\chi(t)$ for different values of the SNR. As the SNR increases, the recovery of $\chi(t)$ improves. To illustrate the most general case where $\chi(t)$ is not known to be monotonic we do not apply the monotonicity check (step 6 of Fig. 1).

where $\star$ stands for the discrete form of the convolution operation (1) used to predict the responses, i.e. $\Delta t \sum_i \chi^0_{j-i} f^k_i$. In the

following we denote $\varepsilon^0_k$ the *prediction error* for the experiment "$k$". To measure the quality of the prediction across multiple experiments we define also the *mean prediction error*

$$\varepsilon^0 := \frac{1}{K} \sum_{k=1}^{K} \varepsilon^0_k, \tag{37}$$

where $K$ is the number of predicted responses. The reader may wonder why we quantify the quality of the recovery only indirectly from the responses found in different experiments and not directly from the recovery of $\chi(t)$. The reason is that in

real applications the "true" $\chi(t)$ is not known but the responses are. The reliability of this indirect measure for the quality of the recovery is discussed in section 5.

To study how the quality of the recovery depends on the noise level we introduce the signal-to-noise ratio (SNR) of the response data from a perturbation experiment as

$$SNR := \frac{||\boldsymbol{\Delta Y}||}{\delta}, \tag{38}$$

where $\delta$ is the final noise level estimate obtained by the RFI method, as described in section 3.4 (see Eq. (29)).

To demonstrate the dependence of the mean prediction error (37) on the SNR, we performed 1% experiments using different noise levels. The resulting dependence is shown in Fig. 4. As expected, for a small error a sufficiently large SNR is needed, i.e. a good recovery may be hindered by a too large noise level.

In Fig. 5 we demonstrate how the overall noise level adjustment in step 3 of the RFI algorithm (see Fig. 1) affects regular-

ization to recover the correct response function. To guarantee that the overall level of the noise spectrum is indeed substantially





different in the control and perturbed experiment (so that the adjustment is really needed), we take for the noise in the control experiment a standard deviation ten times smaller than that for the noise in the perturbed experiment. To demonstrate how the adjustment works it is helpful to consider the so-called "Picard plot". This type of plot was originally introduced to analyze the spectral characteristics of an ill-posed problem (see e.g. Hansen, 1992). In Fig. 5(a) we show the Picard plot for data obtained

from a 1% experiment with the toy model using a SNR $\approx$ 520 to assure a good recovery. The singular values $\sigma_i$ decrease to extremely small values as the index $i$ increases. This demonstrates that indeed the problem to solve for the response function is ill-posed and therefore regularization is needed for its solution (compare Eq. (19) with Eq. (20)). The data labelled by $|\boldsymbol{u}_i \bullet \boldsymbol{\eta}|$ are the "true" noise coefficients, obtained by subtracting the "clean" response $\mathbf{A}\boldsymbol{q}$, known analytically from the toy model description, from the noisy toy model response $\boldsymbol{\Delta Y}$. Comparing them to the projection coefficients of the response $|\boldsymbol{u}_i \bullet \boldsymbol{\Delta Y}|$

one sees that with exception of the first few coefficients the response is dominated by its noise content. Accordingly, only the information contained in these first few coefficients is recoverable from this ill-posed problem whatever method is used. The data labelled by $|\boldsymbol{u}_i \bullet \boldsymbol{\eta}_{est}|$ have been added to the Picard plot to demonstrate how the RFI algorithm operates: These data are the projection coefficients of the estimated noise content in the data, where $\boldsymbol{\eta}_{est}$ is the final value of $\boldsymbol{\eta}'$ obtained by the RFI method. Obviously, the RFI algorithm correctly estimates the "true" noise level not only at high frequencies – where it

is correct by the noise level adjustment in step 3 of the RFI algorithm (see Fig. 1) – but also at low frequencies, where it is predicted from the adjusted low-frequency components of the control experiment (also step 3). Accordingly, in this case the spectral similarity assumption holds and there is no need to further adjust the noise level (step 6).

How the estimation of the noise in the data and the resulting regularization affects the projection coefficients of the spectrum $\boldsymbol{q}$ can be seen in Fig. 5(b): Only those few coefficients not dominated by noise contribute to the regularized solution. In this

case these few coefficients selected by determinining the regularization parameter $\lambda$ from the noise level are sufficient for an almost perfect recovery of the response function, as seen in Fig. 5(c).

It is important to note that in the situation of Fig. 5 where the overall noise level differs considerably in the control and in the perturbed experiment, a naive noise estimate taken from the control experiment without the adjustment in step 3 (as first suggested in section 3.4) would severely underestimate the noise actually in the data. This would in turn lead to an underesti-

mation of the regularization parameter (see Groetsch, 1984, Theorem 3.3.1). As a result, the wrong filtering by regularization would leave projection coefficients dominated by noise in the solution, likely leading to large errors in the recovered response function. This example therefore demonstrates the relevance of the noise adjustment in step 3.

Finally in this section, we demonstrate that by accounting for monotonicity of the linear response function one may obtain a better estimate of the low-frequency components of the noise whereby the recovery of the response function is improved. In

Fig. 6 we plot results from toy model experiments where the spectral similarity assumption does not hold. This was achieved by artificially enhancing the low-frequency components of the noise $\eta^*(t)$ in Eq. (32). The top row plots show the results from the recovery when the additional noise level adjustment was not used. Because the spectral similarity assumption does not hold, the estimated low-frequency components of the noise $|\boldsymbol{u}_i \bullet \boldsymbol{\eta}_{est}|$ do not match those of the "true" noise $|\boldsymbol{u}_i \bullet \boldsymbol{\eta}|$ (subfigure ($a_1$)). Ideally, only those four projection coefficients of the data $|\boldsymbol{u}_i \bullet \boldsymbol{\Delta Y}|$ which are larger than the projection coefficients of the

"true" noise $|\boldsymbol{u}_i \bullet \boldsymbol{\eta}|$ should contribute to the recovered response function. Instead, as seen in subfigure ($b_1$), the coefficients with

**Figure 5.** Demonstration of the operation of the RFI algorithm in the presence of noise using toy model data from a 1% and a control experiment. To demonstrate the relevance of the noise level adjustment (step 3 from Fig. 1), the standard deviation of the noise in the control experiment was taken ten times smaller than that for the noise in the perturbed experiment. (a) Picard plot showing the singular values $\sigma_i$ and the projection coefficients of the data $|\boldsymbol{u}_i \bullet \boldsymbol{\Delta Y}|$, the "true" noise $|\boldsymbol{u}_i \bullet \boldsymbol{\eta}|$, and the final noise estimate $|\boldsymbol{u}_i \bullet \boldsymbol{\eta}_{est}|$; (b) coefficients of regularized solution (20); (c) "true" and recovered linear response functions. Since the RFI algorithm correctly adjusted the noise level to the "true" noise in the data, the resulting regularized solution has contributions only from the first few projection coefficients which are not completely obscured by noise. Overall, the recovery is almost perfect, because the SNR (chosen as about 520) is still sufficiently good and because the noise was chosen to conform with the spectral similarity assumption. The regularization parameter determined by the algorithm is $\lambda \approx 30364$. Because the noise level adjustment (step 3 from Fig. 1) already gave a good estimate to the "true" noise in the data, no monotonicity check was needed (step 6 from Fig. 1).

index between $i = 4$ and $i = 7$ give the dominant contributions because they are larger than the estimated noise coefficients $|\boldsymbol{u}_i \bullet \boldsymbol{\eta}_{est}|$ (compare subfigure ($a_1$)). Therefore, the recovery of the response function is poor (subfigure ($c_1$)). But since in





this case the low-frequency components of noise are such that the recovered response function is non-monotonic although the "true" response function is known to be monotonic, one may further adjust the noise level to improve the results.

This further adjustment is the purpose of step 6 of the RFI algorithm (see Fig. 1). Its effect is demonstrated by the second-row plots of Fig. 6: The estimated noise components match now better the "true" noise components that had been underestimated in the first row (compare subfigures ($a_2$) and ($a_1$)) so that only those four components that carry information (compare in subfigure ($a_2$) the projections $|\boldsymbol{u}_i \bullet \boldsymbol{\Delta Y}|$ for low index $i$ with $|\boldsymbol{u}_i \bullet \boldsymbol{\eta}|$) survive the regularization (subfigure ($b_2$)). As a result, the quality of the recovery of the response function has considerably improved (subfigure ($c_2$)).

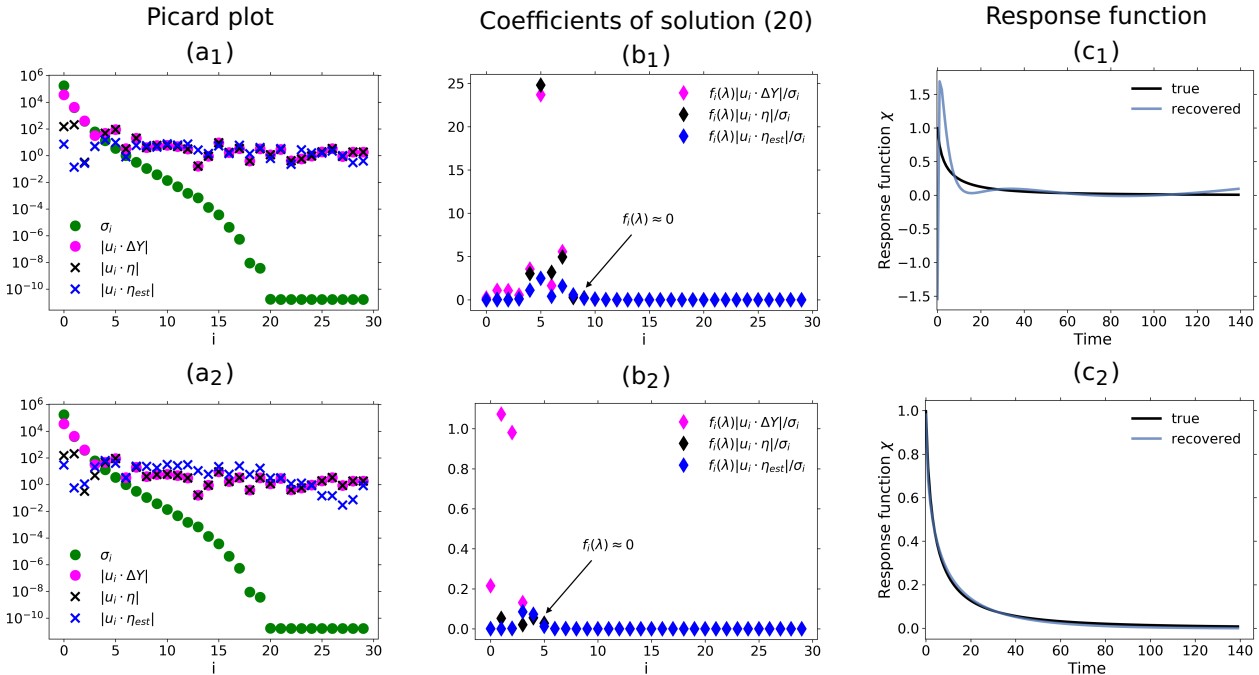

**Figure 6.** Demonstration of the additional noise level adjustment in the presence of a monotonicity constraint using toy model data from a 1% and a control experiment: (a) Picard plot; (b) coefficients of regularized solution (20) and (c) recovered linear response function. All figures are based on the same toy model experiments using a SNR = 1189. To demonstrate the effect of the noise level adjustment the spectral similarity assumption is broken by artificially increasing the low-frequency components of the noise in the experiments. The plots in the first row show the results from the RFI algorithm in the absence of additional noise level adjustment (step 6 in Fig. 1). Although the "true" response function of the toy model is monotonic, the response function recovered by the RFI algorithm is non-monotonic (last figure in the first row). But if the noise adjustment is switched on (second row), the response function is correctly recovered as monotonic (last figure in the second row). Arrows in subfigures (b) indicate the index $i_{critical}$ that separates components of the solution that are only weakly suppressed ($i < i_{critical}$) from those that are almost completely suppressed ($i \geq i_{critical}$). The regularization parameter determined by the algorithm is $\lambda \approx 1$ for the first row and $\lambda \approx 11450$ for the second. For more details see text.





## 4.5 Second complication: nonlinearity

The second difficulty in recovering the linear response function $\chi(t)$ from a perturbation experiment may arise from nonlinearities present in the considered system. Generally it must be suspected that nonlinearities are present so that they should not hurt as long as they are small. And indeed, from the viewpoint of regularization, contributions from nonlinearities can be considered as an additional noise so that in principle they can also be filtered out. But as with noise, when getting stronger they cause a deterioration of the recovery of the response function. In the following, we show this more formally and discuss in detail how the RFI algorithm behaves in the presence of nonlinearities.

To understand how contributions from nonlinearities affect the recovery of the response function we write the nonlinear terms in Eq. (5) collectively as $\widetilde{\eta}(t)$. This formally gives

$$\mathbf{\Delta Y} = \mathbf{A}\boldsymbol{q} + \boldsymbol{\eta} + \widetilde{\boldsymbol{\eta}} \tag{39}$$

instead of Eq. (13). Plugging this into Eq. (20) the spectrum is obtained as

$$\boldsymbol{q}_\lambda = \sum_{i=0}^{M-1} f_i(\lambda) \left( \frac{\boldsymbol{u}_i \bullet \mathbf{A}\boldsymbol{q}}{\sigma_i} \boldsymbol{v}_i + \frac{\boldsymbol{u}_i \bullet (\boldsymbol{\eta} + \widetilde{\boldsymbol{\eta}})}{\sigma_i} \boldsymbol{v}_i \right). \tag{40}$$

Accordingly, the nonlinear contributions can be understood as an additional noise in the spectrum $\boldsymbol{q}_\lambda$ so that the theory of regularization fully applies when replacing $\boldsymbol{\eta}$ by the *combined noise* $\boldsymbol{\eta} + \widetilde{\boldsymbol{\eta}}$. Hence, as in their absence, nonlinearities do not prevent the application of regularization as long as the signal is not buried under this combined noise.

But for the RFI algorithm to give good results a second condition is that the contributions from $\widetilde{\boldsymbol{\eta}}$ must not be large compared to those from $\boldsymbol{\eta}$. To understand this, one must realize that the response and with it the nonlinear contributions $\widetilde{\boldsymbol{\eta}}$ are dominated by low-frequency components because of the low-frequency nature of the forcing for the problems of interest (for instance in %-experiments). The RFI algorithm uses an estimate for the noise level in the perturbation experiment obtained from the control experiment assuming that the spectral distribution is approximately the same in the noise from the control experiment and the noise in the data from the perturbation experiment (spectral similarity assumption; step 3 of Fig. 1). But the control experiment does not contain any contributions from nonlinearities because the forcing is zero. Therefore, if in the data from the perturbation experiment the contributions from nonlinearities $\widetilde{\boldsymbol{\eta}}$ are not small compared to those from $\boldsymbol{\eta}$, the spectral similarity assumption does not hold. Since this assumption is at the heart of the RFI algorithm, its breakdown leads to a poor recovery of the linear response function.

All this is demonstrated in the following by toy model experiments. For this purpose, we artificially consider the response of the toy model not in $Y$ but in its nonlinear transform

$$Y_{nonlin}(t) := Y(t) - aY^2(t), \tag{41}$$

where the parameter $a$ determines the strength of the nonlinearity. Indeed, in such a way the nonlinearity does not result from nonlinearity of the underlying dynamics (the toy model is linear), but from the way the response is looked at. But this distinction is artificial since in practice a response experiment is an indivisible unity of system and observation so that the origin


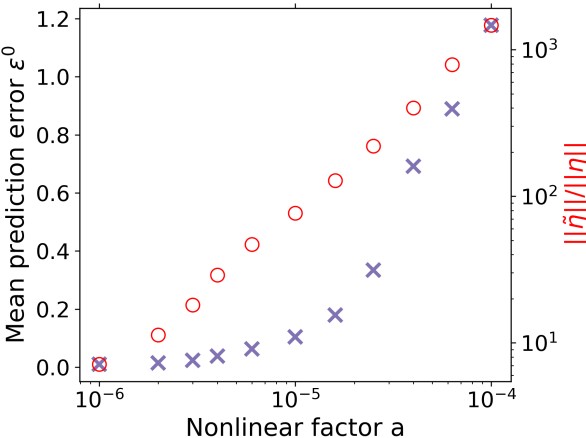

**Figure 7.** Mean prediction error (37) of the recovery when deriving $\chi(t)$ for different values of the nonlinearity factor $a$ of the toy model. As $a$ increases, the recovery of $\chi(t)$ deteriorates because the level of the contributions from nonlinearities $||\widetilde{\boldsymbol{\eta}}||$ gets large compared to the noise level $||\boldsymbol{\eta}||$; how these terms are computed for the toy model is explained in Appendix D. To demonstrate here the pure effect from the breakdown of the spectral similarity assumption, the RFI algorithm is used here without the additional noise level adjustment enforcing monotonicity.

of the nonlinearity is irrelevant. The particular functional form chosen for $Y_{nonlin}(t)$ mimics the nonlinear effect of saturation encountered for instance in the land carbon sink when atmospheric $CO_2$ rises to high values. In the following, to demonstrate the effect of nonlinearities, we set the noise level in the toy model experiments to a rather small value in order to have a good $SNR$ in the experiments considered.

560    In Fig. 7 we show by plotting the mean prediction error (see Eq. (37)) how the recovery of the response function deteriorates as the nonlinearity parameter $a$ increases. To demonstrate that this is indeed caused by a breakdown of the spectral similarity assumption we plot in addition the ratio $||\widetilde{\boldsymbol{\eta}}||/||\boldsymbol{\eta}||$. It is seen that indeed, as claimed above, the recovery works well only when this ratio is not large, i.e. when the contributions from nonlinearities $\widetilde{\boldsymbol{\eta}}$ are not large compared to those from the noise $\boldsymbol{\eta}$.

    More insight into how nonlinearities affect the recovery is obtained from the more detailed SVD analysis shown in Fig. 8.

565  The first row of subfigures was obtained from the toy model assuming a rather small nonlinearity ($a = 10^{-10}$). In the Picard plot (subfigure $(a_1)$) it is seen that in this case both conditions necessary for a good recovery are met: First, the signal $|\boldsymbol{u}_i \bullet \boldsymbol{\Delta Y}|$ is well visible above the combined noise $|\boldsymbol{u}_i \bullet (\boldsymbol{\eta} + \widetilde{\boldsymbol{\eta}})|$ (see the first four components). Second, in this case $|\boldsymbol{u}_i \bullet \widetilde{\boldsymbol{\eta}}|/|\boldsymbol{u}_i \bullet \boldsymbol{\eta}|$ is small over the whole spectrum, i.e. the contributions from $\widetilde{\boldsymbol{\eta}}$ are small compared to those from $\boldsymbol{\eta}$. As explained above, because this second condition is also met, the noise estimate from the RFI algorithm $\boldsymbol{\eta}_{est}$ is a good approximation to the

570  combined noise across all frequencies (compare in the Picard plot $|\boldsymbol{u}_i \bullet (\boldsymbol{\eta} + \widetilde{\boldsymbol{\eta}})|$ to $|\boldsymbol{u}_i \bullet \boldsymbol{\eta}_{est}|$). As a result, the four components selected by the regularization for the recovered solution (subfigure $(b_1)$) are precisely those dominated by the signal (compare $f_i(\lambda)|\boldsymbol{u}_i \bullet \boldsymbol{\Delta Y}|/\sigma_i$ with $f_i(\lambda)|\boldsymbol{u}_i \bullet (\boldsymbol{\eta} + \widetilde{\boldsymbol{\eta}})|/\sigma_i$). This example demonstrates that as long as these two conditions are met, small contributions from nonlinearities do not prevent a good recovery of the response function (see subfigure $(c_1)$).




In the second row of Fig. 8, we demonstrate how the violation of the second condition obstructs the recovery. In this case
the nonlinearity parameter has been given a larger value ($a = 2.5 \times 10^{-5}$). As a consequence, one sees in the Picard plot that
the low-frequency components of the combined noise are enhanced. The first condition is still met: The signal $|\boldsymbol{u}_i \bullet \boldsymbol{\Delta Y}|$ is
visible above the combined noise $|\boldsymbol{u}_i \bullet (\boldsymbol{\eta} + \widetilde{\boldsymbol{\eta}})|$ (see the first two components). But now the ratio $|\boldsymbol{u}_i \bullet \widetilde{\boldsymbol{\eta}}|/|\boldsymbol{u}_i \bullet \boldsymbol{\eta}|$ gets large
at low frequencies, violating the second condition. As explained, the violation of the second condition leads to the breakdown
of the spectral similarity assumption. As a result, the RFI algorithm underestimates the combined noise at low frequencies
(compare in the Picard plot $|\boldsymbol{u}_i \bullet (\boldsymbol{\eta} + \widetilde{\boldsymbol{\eta}})|$ to $|\boldsymbol{u}_i \bullet \boldsymbol{\eta}_{est}|$). Using this wrong noise estimate, regularization selects components for
the recovered solution that are to a large extent dominated by the combined noise (see components $i = 2$ to $i = 6$ in subfigure
($b_2$)). The result is that the strong low-frequency contributions from nonlinearities deteriorate the recovery of the response
function at long time scales (subfigure ($c_2$)).

In the third row, we demonstrate for this type of nonlinearity that by accounting for monotonicity one can remove from the
recovered solution all components dominated by noise. For this purpose, we set the nonlinearity parameter to the same value as
for the second row ($a = 2.5 \times 10^{-5}$) but employ the additional noise level adjustment (step 6 of Fig. 1), i.e. the low-frequency
range of the noise estimate is now automatically adjusted in order to recover a response function that decays monotonically to
zero. As seen in the Picard plot, the additional noise level adjustment results in an artificial enhancement of the low-frequency
components of the noise estimate, with a large jump separating the low- from the high-frequency range. In this case, such
enhancement is able to better estimate the largest components of the combined noise (first few components in the Picard plot).
As a consequence, regularization correctly selects for the recovered solution only the two first components which are not
dominated by noise (subfigure ($b_3$)). Unfortunately, as seen in subfigure ($c_3$), these two first components do not contain enough
information for a perfect recovery, since the quality improves at long time scales, but deteriorates at short time scales (compare
subfigures ($c_3$) and ($c_2$)). This is a consequence of how regularization works: It filters out components dominated by noise (or
in this case, nonlinearity) at the expense of removing also useful information contained in those components.

Also interesting to note from this SVD analysis is that although in general the presence of nonlinearities cannot be detected
from only the two experiments needed for our RFI method, it can be detected in cases where the response function is known
to be monotonic but the nonlinearity is such that the recovered response function is non-monotonic. This is shown in the
last example above, where strong nonlinearities result in a large jump between the low- and high-frequency components of
the noise estimate. Such jump arises because strong nonlinearities cause the response function to be non-monotonic, and this
enforces the additional adjustment of the noise estimate by the RFI algorithm. This effect is obviously a result of the particular
type of nonlinearity considered for that example. Nevertheless, such jump may be a relevant indication of strong nonlinearities
in applications to the land carbon cycle because this type of nonlinearitiy mimics precisely the saturation behaviour observed
in the land carbon sink under high values of atmospheric $CO_2$.



**Figure 8.** Demonstration of how nonlinearities affect the recovery of the response function: (a) Picard plot; (b) coefficients of regularized solution (20) and (c) recovered linear response function. First row: Nonlinearity factor $a = 10^{-10}$ (no monotonicity check); Second row: Nonlinearity factor $a = 2.5 \times 10^{-4}$ (no monotonicity check); Third row: Nonlinearity factor $a = 2.5 \times 10^{-4}$ (with monotonicity check). The noise is overestimated in the low-frequency spectrum in the third row because nonlinearities yield a derived $\chi(t)$ that does not obey the monotonicity constraint. As a consequence, the method increases the level of low-frequency components until the monotonicity constraint is obeyed. The failure to obey the monotonicity constraint and consequent large overestimation of noise in this case can be taken as an indication of the presence of nonlinearities in the response. Note that the "true" linear response function in this nonlinear case $a \neq 0$ is obtained analytically from the linear case $a = 0$ via Eq. (41) (see Appendix D). The regularization parameter determined by the algorithm is $\lambda \approx 3120$ for the first row, $\lambda \approx 74$ for the second, and $\lambda \approx 14611873$ for the third. For more details see text.



## 5 Comparison with previous methods

As a last test of the quality of the results given by the RFI method in application to the toy model, in this section we compare our method against two existent methods in the literature to identify response functions in the time domain. The comparison is performed for the particular case where the response function is known to be monotonic and also for the more general case where it is not. As a side issue this section reveals also some insight into the relation between the quality of the recovery of $\chi(t)$ as measured by the prediction of responses, and the quality of the recovery of $\chi(t)$ itself.

In climate science, the most commonly used method is to obtain $\chi(t)$ from an impulse response, i.e. the response to a perturbation of Dirac delta-type (e.g., Siegenthaler and Oeschger, 1978; Maier-Reimer and Hasselmann, 1987; Joos and Bruno, 1996; Joos et al., 1996; Thompson and Randerson, 1999; Joos et al., 2013). We call it here *pulse method*. Although this method is conceptually straightforward, in some cases it might not yield satisfactory results. Since the perturbation is only one "pulse", depending on the observable of interest it may give a response with small SNR. As a consequence, the recovered response function may be severely affected by noise. On the other hand, if the strength of the pulse is made large to obtain a good SNR, the linear regime may be exceeded. In this case, the impulse response does not correspond anymore to the linear response function.

The second method consists of deriving the linear response function from a step response, i.e. the response to a Heaviside-type perturbation (e.g., Hasselmann et al., 1993; Ragone et al., 2016; MacMartin and Kravitz, 2016; Lucarini et al., 2017; Van Zalinge et al., 2017; Aengenheyster et al., 2018). We call it here *step method*. Due to the special form of this "step" perturbation, the linear response function can in principle be derived from

$$\chi(t) = \frac{1}{f_{step}} \frac{d}{dt} \Delta Y_{step}(t), \tag{42}$$

where $f_{step}$ is the step perturbation and $\Delta Y_{step}$ is the corresponding response. Unfortunately, such derivation involves numerical differentiation, which is known to be an ill-posed problem (Anderssen and Bloomfield, 1974; Engl et al., 1996). Because the problem is ill-posed, noise is amplified, potentially resulting in large errors in the derived linear response function.

These two methods therefore share two limitations: First, they require a special perturbation experiment; second, because of noise in the data they might yield a response function with large errors. In principle, the second limitation may be overcome by using instead of a single response the ensemble average over multiple responses. But this comes at the expense of the numerical burden of performing multiple experiments, which is especially large when dealing with complex models such as state-of-the-art Earth System Models.

The main advantages of the RFI method lie precisely in overcoming these two limitations: It recovers the response function from any type of perturbation experiment and automatically filters out the noise by regularization.

For the results of this section, we performed ensembles of 200 simulation experiments with the toy model (see section 4.1). Each ensemble member is defined by a realization of the noise $\eta^*(t)$ with a fixed standard deviation (see Eq. (34)). Each realization was added via Eq. (32) to three experiments: 1%, step ($2 \times f_0$), and pulse ($4 \times f_0$). Note that because of the issue with the SNR mentioned above, we had to employ for the pulse experiment twice the forcing strength employed for the step





experiment. Further, for each ensemble member an additional realization of the noise was generated to serve as a control experiment to compute the noise estimate for the RFI method (step 1 of Fig. 1).

We computed the response function by the pulse and step method as follows. For a pulse experiment the forcing is $f(t) = a\delta(t)$ with forcing strength $a$, so that the response is given by

$$\Delta Y_{pulse}(t) = \int\limits_{0}^{t} \chi(t-s)a\delta(s)ds = a\chi(t). \tag{43}$$

Therefore, for the pulse method we took the response from the pulse experiment and obtained the response function by

$$\chi(t) = \frac{1}{a}\Delta Y_{pulse}(t). \tag{44}$$

The recovery by the step method was calculated by taking the response from the step experiment and applying Eq. (42). The derivative was computed by forward difference.

To obtain comparable results with these two methods, we recovered the response function by the RFI method from the same pulse and step experiments. To compare the quality of the results using also an experiment not decidedly tailored for the identification, we include additionally the recovery from the 1% experiment.

To obtain a quantitative comparison for the quality of the recovery for each method, we define the recovery error

$$\varepsilon^r := \frac{||\boldsymbol{\chi} - \boldsymbol{\chi}^*||}{||\boldsymbol{\chi}^*||}, \tag{45}$$

where $\boldsymbol{\chi}$ is the recovered response function and $\boldsymbol{\chi}^*$ is the "true" response function, which is known because we use the toy model. In contrast to the prediction error, that measures the quality of the recovery of $\chi(t)$ by means of the response (see Eq. (45)), the recovery error $\varepsilon^r$ measures the quality of the recovery of $\chi(t)$ itself. Another reason for introducing the recovery

error is to compare its results with results from the prediction error. By doing that, we can gain insight into how much the prediction error can be trusted as an indirect measure of the quality of recovery in real applications, where the "true" response function is not known.

First, we compare the pulse and step methods against the full RFI algorithm, i.e. the RFI algorithm taking monotonicity into account (step 6 in Fig. 1). Results are shown in Fig. 9. In the first row of subfigures, we took for the recovery the ensemble

average over the 200 responses for each experiment. For the RFI method, we took the ensemble average over the control experiments as well to estimate the noise (step 1 of Fig. 1). As shown in subfigure ($a_1$), with this approach all methods recover the response function almost perfectly. The quality of the recovery is quantified by the recovery error in subfigure ($b_1$). The RFI method shows the smallest values for the step and pulse experiments when compared to the step and pulse methods. Overall, the step method clearly shows the largest value. To quantify the quality of the prediction, we plot in subfigure ($c_1$) the

prediction error (36). As seen, values are even smaller than for the recovery error. Overall, we see a similar pattern: the step method again stands out, with other methods showing much smaller error values.

In the second row, we compare results by taking only a single response for the recovery. Since the quality of the recovery by the different methods may vary depending on the particular noise realization, we again performed 200 simulations to obtain


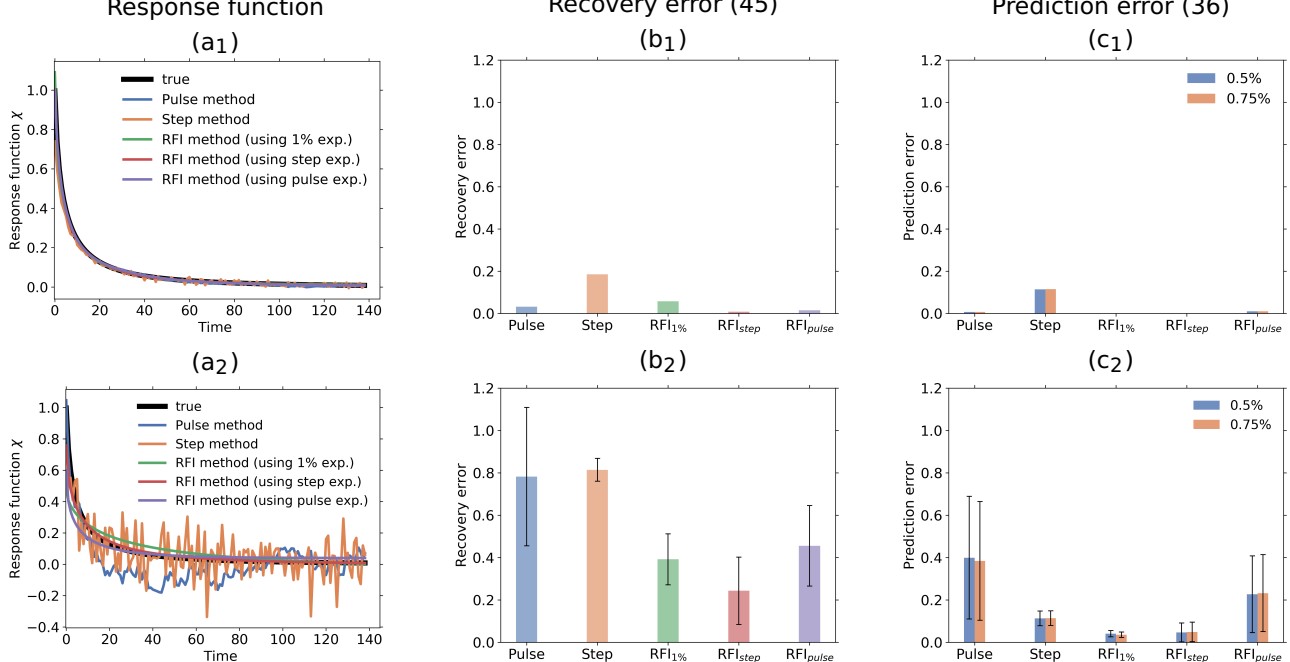

**Figure 9.** Quality of response function recovery by the full RFI method (including step 6 in Fig. 1) in comparison to the pulse and step method. Subscripts at "RFI" indicate the experiment from which the response function was recovered with the RFI method. First row: taking the average over the whole ensemble of toy model experiments for recovery; second row: Performing the recovery for each ensemble member separately. ($a_1$) Recovered response function; ($b_1$) recovery error; ($c_1$) prediction error (36); ($a_2$) example of recovered response function from one ensemble member; ($b_2$) statistics of recovery error; ($c_2$) statistics of prediction error (36). The prediction error is separately computed for the 0.5% and 0.75% experiments. Taking the ensemble average, all methods perform well (see first row). But taking only one ensemble member, the RFI algorithm gives better recovery and prediction errors than the pulse and step methods when comparing the same responses (see second row).

better statistics, but this time deriving the linear response function for each ensemble member separately. Subfigure ($a_2$) shows
an example of recovery for one of the ensemble members. As expected, the recoveries by the pulse and step methods largely
deviate from the true response function. For the pulse method, the large errors result from the low SNR of the pulse response:
Even taking twice the forcing strength of the step experiment, the SNR of the pulse response is of order $10^0$ against order
$10^1$ for the step and 1% responses. For the step method, on the other hand, the large errors are not a result of low SNR, but
of the noise amplification associated with the ill-posedness of numerical differentiation. In contrast to the recovery by these
two methods, because of regularization the recoveries by the RFI method are smoother and visually seem to better fit the true
response function. To quantitatively check these results, we plot in subfigure ($b_2$) for each method the average and standard
deviation over the 200 values of the recovery error (one for each ensemble member). The figure shows that indeed the pulse
and step methods display the largest average recovery error, with the pulse method having a much larger spread. Such spread





is probably related to the low SNR in the response from the pulse experiment. The results from the 1% and pulse experiments by the RFI method are better, showing comparable error magnitudes. The smallest average recovery error is obtained from the RFI method using the step experiment. In subfigure $(c_2)$ we show the average and standard deviation over the 200 values of the prediction error (36). The smallest average prediction errors are obtained from the RFI method using the 1% and step experiments. The largest errors are obtained for the pulse method and the RFI method using the pulse experiment. In contrast to the situation for the recovery error, for the prediction error no substantial difference between the two is found. Note also that when comparing recoveries from the same response (i.e. comparing "Pulse" with "$\text{RFI}_{pulse}$" and "Step" with "$\text{RFI}_{step}$"), the RFI method gives better results than both the pulse and step methods. Another interesting point is that prediction errors for the step method remain approximately unchanged by taking the ensemble mean and a single response (compare "Step" in subfigures $(c_1)$ and $(c_2)$). Overall, as in the first row, the prediction error shows for each individual method values smaller than the recovery error. But now there is a difference between the plots for the recovery and prediction error: Although the pulse and step methods show the largest averages with values of comparable size for the recovery error, for the prediction error the pulse method has the largest average with a value much larger than the step method.

This difference can be better understood as follows (see MacMartin and Kravitz (2016) for more details including the influence of the forcing scenario). Because Eq. (1) is ill-posed, the convolution operator acts on $\chi(t)$ as a "low-pass filter" (see e.g. Bertero et al., 1995; Istratov and Vyvenko, 1999). This means that high frequencies in $\chi(t)$ are suppressed by convolution and show up damped in the response $\Delta Y(t)$. Hence, recoveries with large errors only at high frequencies tend to give relatively small prediction errors. Because of the low SNR, the pulse method yields a recovery of $\chi(t)$ with large errors both at high and low frequencies. Although the errors at high frequencies are damped in the prediction, errors at low frequencies are not. Hence, the large recovery error results in a large prediction error. On the other hand, because of the good SNR for the step response, the step method gives a relatively good recovery of $\chi(t)$ at low frequencies, with large errors concentrated at high frequencies. As a result, the large recovery error results only in a small prediction error. This suppression of high-frequency errors might also explain why the prediction error for the step method remains unchanged when recovering the response function from a single response instead of the ensemble average. By comparing the recovery of $\chi(t)$ by the step method in Fig. 9($a_1$) and ($a_2$), one sees that the main difference is indeed at high frequencies (the recovery in Fig. 9($a_2$) is quite "noisy" but follows the long term trend). This is because the noise amplification has a larger effect on the recovery from the single response due to its larger noise level. But since low frequencies are well recovered in both cases, the resulting prediction errors are almost the same.

Overall, the analysis of Fig. 9 suggests two main conclusions. First, as expected, the prediction error gives indeed an indication of the quality of the recovery, since good recoveries result in good predictions. But care should be taken when judging the recovery only from the prediction error, because a good prediction does not necessarily imply a good recovery: Due to the ill-posedness, Eq. (1) might damp large high-frequency recovery errors so that they do not show up in the prediction. Nevertheless, from a good prediction error one can still infer a good recovery at low frequencies, because at these frequencies large recovery errors result in large prediction errors. Since regularization filtering leaves only low-frequency terms in the recovery, the RFI method shows in Fig. 9 small prediction errors associated to small recovery errors.





Second, by taking only a single response – and not the ensemble average – the full RFI algorithm gives on average smaller recovery and prediction errors than the pulse and step methods when comparing results obtained from the same experiment.

But the results above cover only the case where the full RFI algorithm is employed. In the following, we analyze also the case where monotonicity is not taken into account. For this purpose, we repeated in all detail the exercise that led to Fig. 9 but did not apply the additional noise level adjustment to enforce monotonicity of the response function. Figure 10(a) shows the results for the recovery error. Once more, the RFI method gives smaller values than the step and pulse methods when comparing the recovery from the same responses. In addition, now the recovery for the RFI method using the step experiment

even improved in comparison to Fig. 9($b_2$). The reason may be related to the numerical check for monotonicity: Depending on the tolerance value that is used to judge whether the recovered response function is monotonic, the additional adjustment might actually overestimate the noise level, leading to slightly worse results.

Yet, the improvement brought by the additional noise level adjustment is clear when looking at the recovery error for the 1% experiment. Compared to Fig. 9($b_2$), the average error increases substantially, and the spread is much larger (see inset for

the whole value). As explained in section 4.4, this deterioration results from cases where the noise in the response is such that the spectral similarity assumption does not hold. Since here the noise estimate resulting from this assumption is not further improved by the monotonicity check, the result is a poor recovery (see subfigure (b) for an example). But because the large errors are mostly at high frequencies, even poor recoveries are still sufficiently good for predictions, as shown by the small mean prediction error in subfigure (c) (see "RFI$_{1\%}$"). Therefore, in contrast to the case where monotonicity is taken into

account, here some small prediction errors are associated to large recovery errors.

Nevertheless, we find that although extreme, such poor recoveries are not frequent. In fact, extreme cases with recovery error $\varepsilon^r > 1$ account for 6.5% of the recoveries. This suggests that the large deterioration in the mean and spread of the recovery error in subfigure (a) is not a result of overall poor recoveries, but of only few extreme cases. To check this hypothesis, we plot in subfigure (d) the mean and standard deviation excluding these cases from the calculations. Indeed the result is much

better, showing values comparable to the case where monotonicity is taken into account (compare "RFI$_{1\%}$" in Fig. 10(d) and Fig. 9($b_2$)). Overall, this result indicates that at least for models of this type – where in the perturbation experiment the spectral distribution of noise does not change drastically compared to the control experiment – although monotonicity plays a role in avoiding large recovery errors, statistically most recoveries are still relatively good even without this additional improvement.

## 6 Summary, discussion and outlook

Existent methods to identify linear response functions from data require tailored perturbation experiments. Here, we developed a method to identify linear response functions from data using only information from an arbitrary perturbation experiment and a control experiment. The RFI method adresses the ill-posedness inherent to the identification problem by applying Tikhonov-Phillips regularization. The regularization parameter is computed by the discrepancy method, which involves the estimation of the noise level. For this purpose, we take advantage of information given by a spectral analysis of the perturbation experiment

and by the control experiment. Assuming that the Picard condition holds, we estimate from the perturbation experiment the


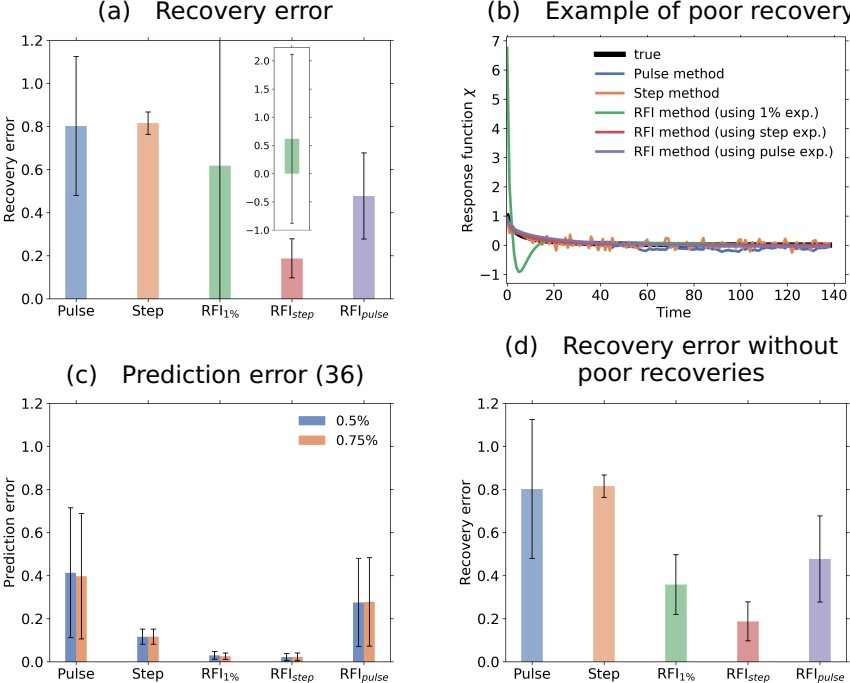

**Figure 10.** Quality of response function recovery by our RFI method excluding step 6 in Fig. 1 in comparison to the pulse and step method. Response function is recovered taking the individual response for each ensemble member. Subscripts at "RFI" indicate the experiment from which the response function was recovered with the RFI method. (a) Statistics of the recovery error; (b) example of poor recovery with the RFI algorithm; (c) statistics of the prediction error (36); (d) statistics of the recovery error excluding for the $RFI_{1\%}$ the 6.5% of the recoveries with recovery error greater than 1. Once again, the RFI method gives better recovery and prediction errors than the Pulse and Step methods for the same responses. Without accounting for monotonicity the variability in the quality of the recoveries from the 1% experiment increases substantially, but poor recoveries are obtained only in few cases.

high-frequency components of the noise. Then, assuming that the spectral distribution of noise is approximately the same for the perturbed and control experiments (spectral similarity assumption), we estimate from the control experiment the low-frequency components of the noise. The obtained noise level estimate can be further adjusted if the linear response function is known to be monotonic. The robustness of the method in the presence of noise and nonlinearity was demonstrated in section 4.

Additional sensitivity tests showing the robustness of the method under changes in the parameters for the recovery are shown in Appendix E.

   As discussed in section 5, the developed method to identify linear response functions is an alternative approach to existent methods in the literature, which require special perturbation experiments and often give results with large errors caused by noise. In contrast, the RFI method accounts in a systematic way for the noise and can be directly applied to data from any type

of perturbation experiment once also a control experiment is given. Because it filters out the noise, its results show in the cases analyzed here a higher quality compared to results from previous methods when applied to the same data from a toy model.





And because it can identify response functions from any type of perturbation experiment, the method is particularly suitable for application to data from the C$^4$MIP carbon cycle model intercomparison as shown in Part II of this study.

The main novelty of the method is the estimation of the noise level (steps 1–3 of Fig. 1), which is known to be critical for the

application of regularization theory. When solving a problem by regularization, the most crucial step is the computation of the regularization parameter. To compute this parameter in a way that the solution converges to the "true" solution for decreasing noise, methods need to account for the noise level (Bakushinskii, 1984; Engl et al., 1996). But in practical applications the noise level is rarely known. Therefore, methods to obtain good estimates are needed. Our new method to estimate the noise level consists essentially of two steps: First, estimating the high-frequency components from data and then the low-frequency

components from the control experiment. While the second step is completely novel, the main idea behind the first step was already brought up in earlier studies (e.g., Hansen, 1990) and has recently been further developed by methods to compute the "Picard parameter" (Taroudaki and O'Leary, 2015; Levin and Meltzer, 2017), which is different from the $i_{max}$ that we use in our method. The Picard parameter is computed as the index for which the components $|\boldsymbol{u}_i \bullet \boldsymbol{\Delta Y}|$ start to level off. Typically, from this index onwards the data can be interpreted as noise. For this reason, one might think that from the Picard parameter

one can obtain all data components dominated by noise and thereby estimate the noise level. But this is not generally true: For instance, if the noise has large low-frequency components such as in Fig. 8(a$_2$), then the components $|\boldsymbol{u}_i \bullet \boldsymbol{\Delta Y}|$ level off at an index larger than that at which the data starts to be dominated by noise, so that in this case the Picard parameter does not determine all data components dominated by noise. In our RFI method, the interest lies in obtaining not all data components dominated by noise, but only enough components to obtain the overall level of the high-frequency noise. For this purpose, we

define instead of the "Picard parameter" the more conservative index $i_{max}$, above which the singular values are zero and by the Picard condition also the "true" data components must be zero. In this way, we unambiguously identify data components that contain only noise (see Eq. (24)). These components give the high-frequency noise level so that in the second step also the remaining low-frequency noise components can be estimated from the control experiment (step 3 of Fig. 1).





Because our noise level estimation is not particularly related to the problem of identifying response functions, it can in principle be applied to solve also other types of linear ill-posed problems (see e.g. Engl et al., 1996). In general, all one needs for the application is:

1. A problem of the type

$$\boldsymbol{y} = \mathbf{A}\boldsymbol{x} + \boldsymbol{\eta}, \tag{46}$$

where given the matrix $\mathbf{A}$ and the noisy data $\boldsymbol{y}$ one is interested in finding $\boldsymbol{x}$.

2. Data from a situation similar to the control experiment, where $\mathbf{A}\boldsymbol{x} = 0$, so that the resulting $\boldsymbol{y}_{ctrl}$ gives the noise term

$$\boldsymbol{\eta}_{ctrl} = \boldsymbol{y}_{ctrl}. \tag{47}$$

3. The singular values of $\mathbf{A}$ decaying to values sufficiently close to zero to obtain $i_{max}$.

Then, as long as both the Picard condition and the spectral similarity assumption hold, the method gives a reasonable noise estimate – since then, by assumption, the noise estimate is simply a scaling of the noise in the control experiment (see section 3.4) – by which the regularization parameter can be determined.

While the Picard condition is necessary for a solution to be recoverable from an ill-posed problem, the validity of the spectral similarity assumption is less clear. An intuitive explanation for this assumption can be thought as follows. Since here the interest lies in identifying linear response functions, the perturbation to the system must be sufficiently weak so that the response can be considered linear. If the noise in the control experiment depends on the perturbation, a sufficiently weak perturbation will modify its characteristics only slightly. The RFI method accounts partially for this change by adjusting the overall level by which the noise increases. Nevertheless, it assumes that since the characteristics of the noise change only slightly, then the spectral components of the noise in the perturbed experiment can be thought as having the same relative contributions as those in the control experiment. When in addition the response function is known to be monotonic, the estimate of the noise can be further improved (step 6 of Fig. 1), this time by adjusting the relative contribution of the spectral components: Since the high-frequency region is known from the spectral analysis of the response, then the components of the noise are adjusted in the low-frequency region; this is done iteratively until the resulting response function gets monotonic. Such additional adjustment has been demonstrated to give good results in the applications in the present study and subsequent Part II for the special case where the response function can be considered monotonic.

Although it is assumed that $\chi(t)$ is given by the spectral form (9), this is not essential for our method. In principle, any functional form can be assumed for $\chi(t)$, or even none – in which case one would recover $\chi(t)$ pointwise. But compared to the simpler pointwise recovery of $\chi(t)$, assuming Eq. (9) has some advantages. The most obvious is that in contrast to the pointwise approach, with Eq. (9) both $\chi(t)$ and the spectrum can be recovered together. If $\chi(t)$ is recovered pointwise, the spectrum has to be derived in a second step from $\chi(t)$, which is also an ill-posed problem (Istratov and Vyvenko, 1999). Further, the description (9) restricts the function space for the recovered $\chi(t)$, forcing $\lim_{t\to\infty} \chi(t) = 0$ as is expected for





most problems of interest, which greatly simplifies the problem compared to the case where $\chi(t)$ can assume any form. Our ansatz (9) has also advantages in comparison with the typical multi-exponential ansatz (7) assumed in most previous studies (see discussion in section 3.1). When assuming that $\chi(t)$ is given by a sum of few exponents, an important problem is how to choose the number of exponents. The typical methods to choose this number rely on "quality-of-fit" criteria; but for ill-posed problems these criteria can be unreliable because in these problems a good fit does not mean that the derived parameters

are close to the "true" parameters (see e.g. the famous example from Lanczos, 1956, p. 272). In our approach, as long as the distribution of time scales is appropriately prescribed and the data quality is sufficiently good, numerical results indicate that the solution is approximately independent of the number of exponents (Appendix E). Moreover, compared to the multi-exponential approach, our ansatz (9) has two additional advantages: The first is that it leads to the linear problem of finding only the spectrum $q(\tau)$ – in contrast to the nonlinear problem of finding both the time scales $\tau_i$ and the weights $g_i$ from Eq. (7)

–, which permits an analytical solution and thereby gives more transparency to the method. The second is that compared to the assumption of only a few time scales, the ansatz of a continuous spectrum of time scales is typically more realistic for real systems, which is e.g. the case for the carbon cycle study presented in Part II. One limitation is however that our ansatz (9) restricts the solution to the class of responses with $\tau$ being real. The most general ansatz would allow complex values for $\tau$. Hence by the ansatz (9) oscillatory contributions to $\chi(t)$ are excluded.

In the present paper the robustness of our method has been investigated only for artificial data taken from toy model experiments. In this analysis, we not only knew the "true" response function underlying the data but also had control over the two complications that may hinder its recovery, namely the level of background noise and nonlinearities. Under these ideal conditions, we could carefully examine the quality of the response functions identified by our RFI method. Nevertheless, such conditions are hardly met in practice. Therefore, the applicability of our method must be investigated as well for real problems.

Such an investigation is presented in Part II of this study.

## Appendix A: Basic equations in this study are Fredholm equations of the first kind

In this appendix we show that Eq. (1), Eq. (7), and Eq. (1) with $\chi(t)$ given by Eq. (7) are indeed special cases of the Fredholm equation of the first kind, as claimed in sections 3.1 and 3.2. Since inverse problems in the form of this equation are well-known to be ill-posed (e.g., Groetsch, 1984; Bertero, 1989; Hansen, 2010), this clarifies the inherent difficulties in identifying linear

response functions from perturbation experiment data.

A Fredholm equation of the first kind is an equation of the type (Groetsch, 1984)

$$h(t) = \int_a^b k(t,s) f(s) ds. \tag{A1}$$

Clearly, by setting $a := 0$, $k(t,s) := 0 \; \forall \; s > t$, and $k(t,s) := k(t-s)$, one obtains the form of Eq. (1) – which can also be seen as a Volterra equation of the first kind (Olshevsky, 1930; Polyanin and Manzhirov, 1998; Groetsch, 2007).





That Eq. (7) is a special case of Eq. (A1) can be seen (Istratov and Vyvenko, 1999) by noting that Eq. (7) can be written in integral form as

$$\chi(t) = \int_0^\infty e^{-t/\tau} g(\tau) d\tau \tag{A2}$$

with

$$g(\tau) = \sum_{i=1}^M g_i \, \delta(\tau - \tau_i). \tag{A3}$$

Since Eq. (A2) is a particular case of Eq. (A1) and Eq. (7) is a particular case of Eq. (A2), Eq. (7) is also a particular case of Eq. (A1).

Now, entering Eq. (7), written in the form (A2)–(A3), into Eq. (1), one obtains an equation of the type

$$R(t) = \int_0^\infty k(t,\tau) g(\tau) d\tau \tag{A4}$$

with

$$k(t,\tau) = \int_0^t e^{-(t-s)/\tau} f(s) ds, \tag{A5}$$

which is a special case of Eq. (A1). Thus, Eq. (1), Eq. (7), and Eq. (1) with $\chi(t)$ given by Eq. (7) can all be understood as Fredholm equations of the first kind.

**Appendix B: Derivation of Eqs. (11) and (12) on which our study is based**

This appendix complements section 3.2 by deriving the set of Eqs. (11), (12) underlying the RFI algorithm. They are a discretization of the basic definition (6) of the linear response function we are interested in. The special form (11), (12) involves in particular the logarithmic transformation (9) and a discretization of the representation (8) for the response function by means of a spectrum of time scales. Since $\chi(t)$ is assumed to be given by a spectrum of time scales according to Eq. (8), the discretization must be performed both in the time and time scale domain.

We start by defining the nondimensional time scale

$$\tau' := \frac{\tau}{\tau_0}, \tag{B1}$$

where $\tau_0$ is a reference time scale. Applying definition (B1) in Eq. (8) gives

$$\chi(t) = \int_0^\infty g(\tau_0 \tau') e^{-t/\tau_0 \tau'} \tau_0 d\tau'. \tag{B2}$$



Due to the wide range of time scales of the systems of interest such as climate and the carbon cycle (Part II of this study), calculations are facilitated if the time scales are evenly distributed at a logarithmic scale. To do so, the following change of variables is performed in Eq. (B2):

$$\tau' = 10^z, \tag{B3}$$

$$d\tau' = 10^z \ln 10\, dz = \tau' \ln 10\, d\log_{10}\tau'. \tag{B4}$$

Thus, Eq. (B2) becomes

$$\chi(t) = \int_{-\infty}^{\infty} g(\tau_0 10^{\log_{10}\tau'}) e^{-t/\tau_0 10^{\log_{10}\tau'}} \tau_0 \tau' \ln 10\, d\log_{10}\tau', \tag{B5}$$

or simply

$$\chi(t) = \int_{-\infty}^{\infty} g(\tau_0 \tau') e^{-t/\tau_0 \tau'} \tau_0 \tau' \ln 10\, d\log_{10}\tau'. \tag{B6}$$

A convenient choice for the reference value is $\tau_0 = 1$ unit of time, so that by Eq. (B1) the time scale $\tau = \tau'$ units of time. The resulting equation can thus be written as

$$\chi(t) = \int_{-\infty}^{\infty} q(\tau') e^{-t/\tau'} d\log_{10}\tau', \tag{B7}$$

with

$$q(\tau') := \tau' \ln 10\, g(\tau'). \tag{B8}$$

For convenience of notation we use simply $\tau$ instead of $\tau'$.

For the discretization the support of $q(\tau)$ is assumed to lie within $[\log\tau_{min}, \log\tau_{max}]$. Accordingly, Eq. (B7) reduces to

$$\chi(t) = \int_{\log\tau_{min}}^{\log\tau_{max}} q(\tau) e^{-t/\tau} d\log_{10}\tau. \tag{B9}$$

Taking a constant step $\Delta\log\tau$ such that $\log\tau_{max} = \log\tau_{min} + M\Delta\log_{10}\tau$, Eq. (B9) may be written as

$$\chi(t) = \sum_{j=0}^{M-1} \int_{\log\tau_{min}+j\Delta\log_{10}\tau}^{\log\tau_{min}+(j+1)\Delta\log_{10}\tau} q(\tau) e^{-t/\tau} d\log_{10}\tau. \tag{B10}$$

Naming $t = (k+1)\Delta t$, Eq. (6) can be rewritten as

$$\Delta Y(t) = \sum_{i=0}^{k} \int_{i\Delta t}^{(i+1)\Delta t} \chi(s) f(t-s) ds + \eta(t). \tag{B11}$$




Plugging Eq. (B10) into Eq. (B11) and rearranging the resulting equation gives

$$\Delta Y(t) = \sum_{j=0}^{M-1} \int_{\log_{10} \tau_{min}+j\Delta \log_{10} \tau}^{\log_{10} \tau_{min}+(j+1)\Delta \log_{10} \tau} K(t,\tau)q(\tau)\, d\log_{10}\tau + \eta(t),$$
(B12)

where

$$K(t,\tau) = \sum_{i=0}^{k} \int_{i\Delta t}^{(i+1)\Delta t} e^{-s/\tau} f(t-s)ds.$$
(B13)

Assuming constant steps $\Delta \log_{10}\tau$ and $\Delta t$ one may apply a quadrature rule (Hansen, 2002) to both Eq. (B12) and Eq. (B13), so that

$$\Delta Y(t) = \Delta \log_{10}\tau \sum_{j=0}^{M-1} K(t,\tau_j)q(\tau_j) + \varepsilon_\tau(t) + \eta(t),$$
(B14)

$$K(t,\tau) = \Delta t \sum_{i=0}^{k} e^{-s_i/\tau} f(t-s_i) + \varepsilon_t(t,\tau),$$
(B15)

where $\varepsilon_\tau(t)$ and $\varepsilon_t(t,\tau)$ are the errors resulting from the discretization. Plugging Eq. (B15) into Eq. (B14) yields

$$\Delta Y(t) \approx \Delta \log_{10}\tau\, \Delta t \sum_{j=0}^{M-1} \widetilde{q}(\tau_j) \sum_{i=0}^{k} e^{-s_i/\tau_j} f(t-s_i) + \eta(t) = \psi(t) + \eta(t),$$
(B16)

where $\widetilde{q}$ is an approximation to $q$ that accounts for the discretization errors. Now, if one requires that $\psi(t_k)+\eta(t_k) = \Delta Y(t_k)$
for particular times $t_k$,

$$\Delta Y(t_k) = \Delta \log_{10}\tau\, \Delta t \sum_{j=0}^{M-1} \widetilde{q}(\tau_j) \sum_{i=0}^{k} e^{-s_i/\tau_j} f(t_k-s_i) + \eta(t_k), \quad k=0,1,...,N-1,$$
(B17)

with the time steps chosen as follows

$$t_k = k\Delta t, \quad k=0,1,...,N-1,$$
(B18)

$$s_i = i\Delta t, \quad i=0,1,...,k,$$
(B19)

and the time scales

$$\tau_j = \tau_{min}10^{j\Delta \log_{10}\tau}, \quad j=0,1,...,M-1.$$
(B20)

In order to simplify the notation, Eq. (B17) is written as

$$\Delta Y_k = \Delta t \sum_{i=0}^{k} \chi_{k-i}\, f_i + \eta_k, \quad k=0,...,N-1,$$
(B21)

$$\chi_k = \Delta \log_{10}\tau \sum_{j=0}^{M-1} q_j e^{-k\Delta t/\tau_j}, \quad k=0,...,N-1.$$
(B22)

These are Eqs. (11) and (12) underlying our study.



**Appendix C: Spectrum $q(\tau)$ positive or negative for all $\tau$ implies $\mathcal{X}(t)$ monotonic**

This appendix is referred to on section 3.5 with the claim that a sufficient condition for $\mathcal{X}(t)$ being monotonic is that all components $q_i$ have the same sign. The proof is as follows.

Let $\mathcal{X}(t)$ be defined by Eq. (9). Then,

$$\frac{d}{dt}\mathcal{X}(t) = -\int_{-\infty}^{\infty} q(\tau)\frac{e^{-t/10^{\log_{10}\tau}}}{10^{\log_{10}\tau}}d\log_{10}\tau. \tag{C1}$$

Since $10^{\log_{10}\tau} \geq 0$, $\frac{e^{-t/10^{\log_{10}\tau}}}{10^{\log_{10}\tau}} \geq 0\ \forall\ t$. Thus, if $q(\tau) \geq 0\ \forall\ \tau$, then $\frac{d}{dt}\mathcal{X}(t) \leq 0\ \forall\ t$. Similarly, if $q(\tau) \leq 0\ \forall\ \tau$, then $\frac{d}{dt}\mathcal{X}(t) \geq 0\ \forall\ t$.

**Appendix D: Response function and noise in the nonlinearized response for the toy model**

In this appendix it is shown how the linear response function and the noise terms are computed in section 4.5 when discussing by means of the toy model the complications arising from nonlinearity. We demonstrate that the linear response function for the nonlinear response (Eq. (41) with $a \neq 0$) of the toy model (section 4.1) can be analytically obtained from the linear case $a = 0$. Additionally, the noise from the control experiment and the combined noise in the response are defined.

We first demonstrate how to obtain the linear response function. Plugging Eq. (32) into Eq. (41) gives

$$Y_{nonlin}(t) = [1 - 2a\eta^*(t)]\int_0^t \mathcal{X}^*(t-s)f(s)ds + \eta^*(t)[1 - a\eta^*(t)] - a\left(\int_0^t \mathcal{X}^*(t-s)f(s)ds\right)^2. \tag{D1}$$

Taking the ensemble average of Eq. (D1) and noting that $\langle\eta^*(t)\rangle = 0$ gives

$$\langle Y_{nonlin}(t)\rangle = \int_0^t \mathcal{X}^*(t-s)f(s)ds + \mathcal{O}(f^2). \tag{D2}$$

Therefore, $\mathcal{X}^*(t)$ obtained for $a > 0$ from the nonlinearized response (41) is the same as for the case $a = 0$.

Now, by taking $f = 0$ in Eq. (D1) one obtains for this nonlinear case the noise from the control experiment:

$$\eta_{ctrl}(t) := \eta^*(t)[1 - a\eta^*(t)]. \tag{D3}$$

To define the combined noise $\eta(t) + \widetilde{\eta}(t)$, one must first define the nonlinear term $\widetilde{\eta}(t)$ from Eq. (39). For the nonlinearized response from the toy model, this term is given by the nonlinear term in Eq. (D1), i.e.

$$\widetilde{\eta}(t) := -a\left(\int_0^t \mathcal{X}^*(t-s)f(s)ds\right)^2. \tag{D4}$$

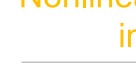
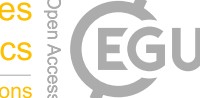

Then, the noise term consists of the remaining terms of the nonlinear response $Y_{nonlin}$ after subtracting the "clean" linear response and the nonlinear term $\widetilde{\eta}$, i.e.

$$\eta(t) := Y_{nonlin}(t) - \int_0^t \chi^*(t-s)f(s)ds - \widetilde{\eta}(t) = -2a\eta^*(t)\int_0^t \chi^*(t-s)f(s)ds + \eta^*(t)[1 - a\eta^*(t)]. \tag{D5}$$

Hence, the combined noise is given by

$$\eta(t) + \widetilde{\eta}(t) := -2a\eta^*(t)\int_0^t \chi^*(t-s)f(s)ds + \eta^*(t)[1 - a\eta^*(t)] - a\left(\int_0^t \chi^*(t-s)f(s)ds\right)^2. \tag{D6}$$

**Appendix E: Sensitivity of the recovered response function and spectrum to the parameters $M$, $\log\tau_{min}$ and $\log\tau_{max}$ of the RFI algorithm**

In this appendix, it is shown that as long as the extent and resolution of the discrete distribution of time scales approximates the spectrum sufficiently densely, the derived spectrum $q_\lambda$ and the derived linear response function $\chi(t)$ are approximately independent of the number of time-scales $M$ and on the limits of the distribution $\log\tau_{min}$ and $\log\tau_{max}$. To isolate the effect of changes in $M$, $\log\tau_{min}$ and $\log\tau_{max}$ from the effect of noise, a relatively high $SNR \sim \mathcal{O}(10^5)$ is taken. For the computations we took data from 1% experiments performed with the toy model described in section 4.1. No monotonicity needed to be accounted for (step 6 of Fig. 1).

Figs. E1–E5 show the recovery taking the same limits used throughout the paper ($\log\tau_{min} = -1$ and $\log\tau_{max} = 5$) but different number of time scales $M$. Figs. E6–E8 show the recovery keeping the number of time scales and the lower limit used throughout the paper ($M = 30$ and $\log\tau_{min} = -1$) but changing the upper limit $\log\tau_{max}$. Figs. E9–E11 show the recovery keeping the number of time scales and the upper limit used throughout the paper ($M = 30$ and $\log\tau_{max} = 5$) but changing the lower limit $\log\tau_{max}$. As expected, the results are approximately independent of the changes in the prescribed parameters. The only substantial differences are found in the recovered spectra at time scales smaller than the time step $\Delta t = 1$, thus time scales over which anyway only little information is given by data. These small time scales are also problematic because of the ill-posedness of the problem that suppresses high-frequency information from the solution (see Groetsch, 1984, section 1.1).

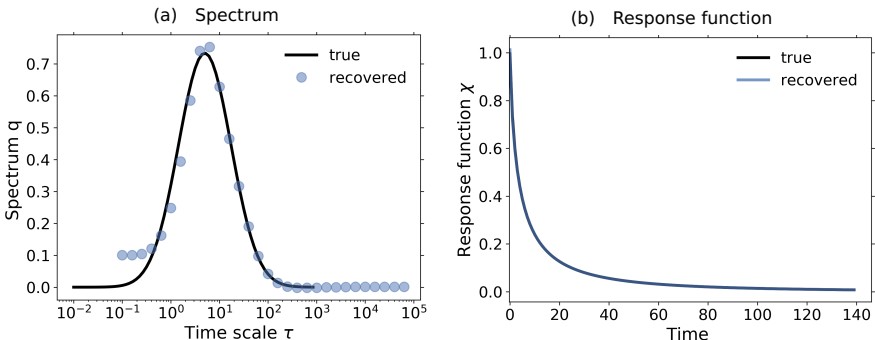

**Figure E1.** Response function $\chi(t)$ and spectrum $q_\lambda$ recovered from toy model data taking the RFI parameters $M = 30$, $\log \tau_{min} = -1$ and $\log \tau_{max} = 5$. Blue dots in (a) and blue line in (b) indicate the recovered values for the spectrum and for the response function, while black lines indicate their "true" values.

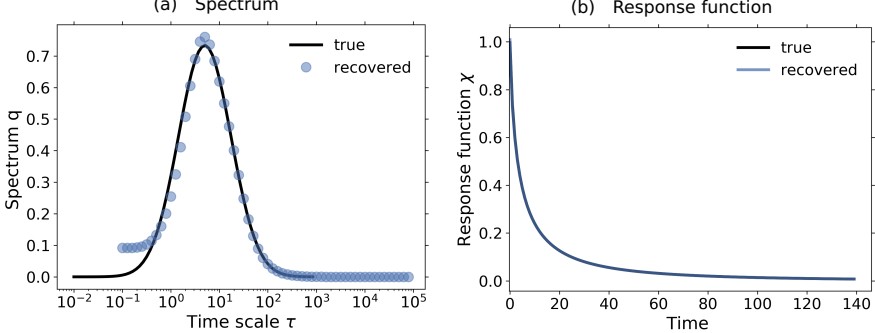

**Figure E2.** Response function $\chi(t)$ and spectrum $q_\lambda$ recovered from toy model data taking the RFI parameters $M = 60$, $\log \tau_{min} = -1$ and $\log \tau_{max} = 5$. Blue dots in (a) and blue line in (b) indicate the recovered values for the spectrum and for the response function, while black lines indicate their "true" values.





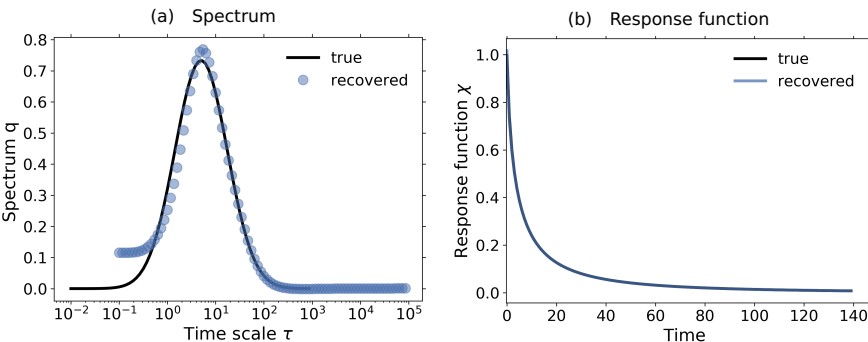

**Figure E3.** Response function $\chi(t)$ and spectrum $q_\lambda$ recovered from toy model data taking the RFI parameters $M = 90$, $\log \tau_{min} = -1$ and $\log \tau_{max} = 5$. Blue dots in (a) and blue line in (b) indicate the recovered values for the spectrum and for the response function, while black lines indicate their "true" values.

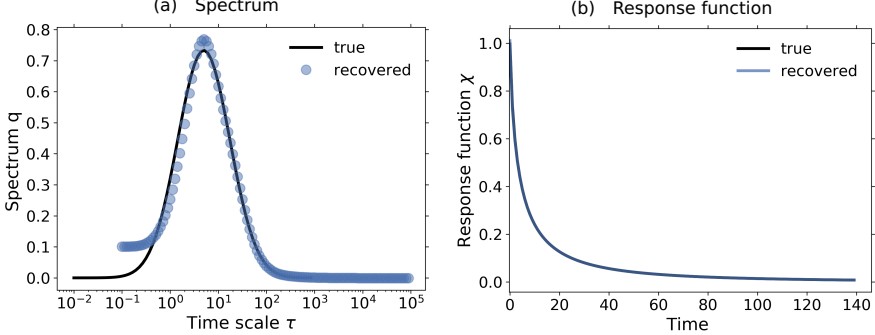

**Figure E4.** Response function $\chi(t)$ and spectrum $q_\lambda$ recovered from toy model data taking the RFI parameters $M = 120$, $\log \tau_{min} = -1$ and $\log \tau_{max} = 5$. Blue dots in (a) and blue line in (b) indicate the recovered values for the spectrum and for the response function, while black lines indicate their "true" values.





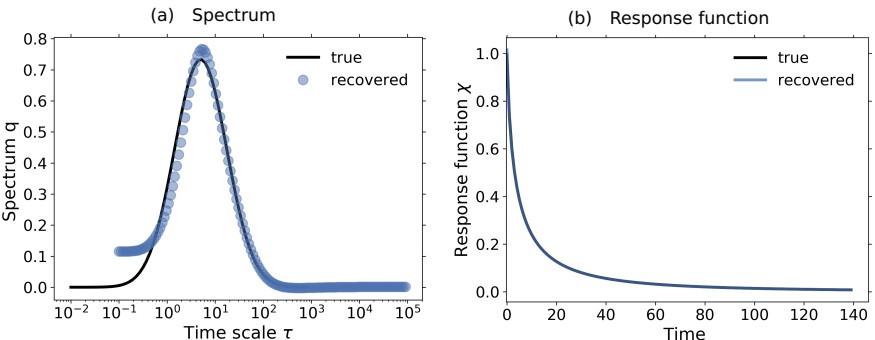

**Figure E5.** Response function $\mathcal{X}(t)$ and spectrum $\boldsymbol{q}_\lambda$ recovered from toy model data taking the RFI parameters $M = 140$, $\log \tau_{min} = -1$ and $\log \tau_{max} = 5$. Blue dots in (a) and blue line in (b) indicate the recovered values for the spectrum and for the response function, while black lines indicate their "true" values.

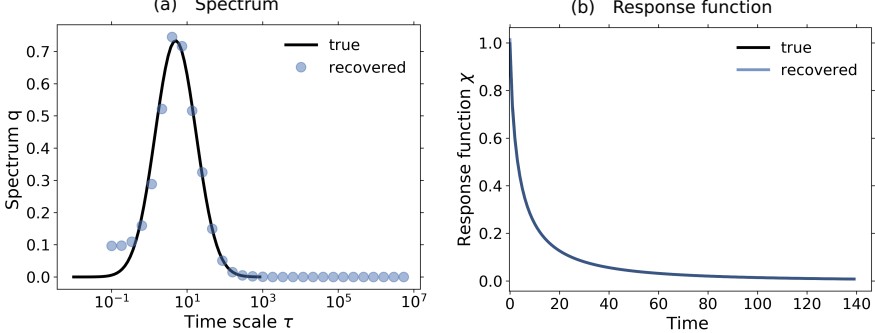

**Figure E6.** Response function $\mathcal{X}(t)$ and spectrum $\boldsymbol{q}_\lambda$ recovered from toy model data taking the RFI parameters $M = 30$, $\log \tau_{min} = -1$ and $\log \tau_{max} = 7$. Blue dots in (a) and blue line in (b) indicate the recovered values for the spectrum and for the response function, while black lines indicate their "true" values.

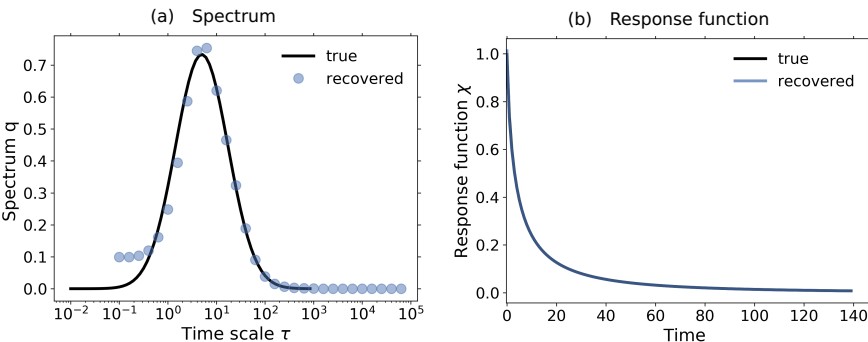

**Figure E7.** Response function $\chi(t)$ and spectrum $\boldsymbol{q}_\lambda$ recovered from toy model data taking the RFI parameters $M = 30$, $\log \tau_{min} = -1$ and $\log \tau_{max} = 5$. Blue dots in (a) and blue line in (b) indicate the recovered values for the spectrum and for the response function, while black lines indicate their "true" values.

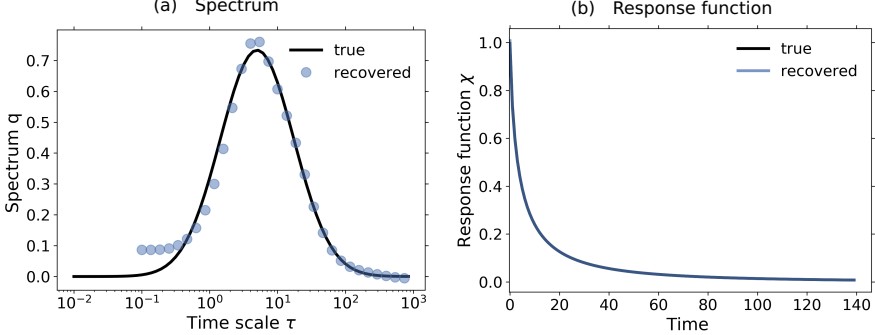

**Figure E8.** Response function $\chi(t)$ and spectrum $\boldsymbol{q}_\lambda$ recovered from toy model data taking the RFI parameters $M = 30$, $\log \tau_{min} = -1$ and $\log \tau_{max} = 3$. Blue dots in (a) and blue line in (b) indicate the recovered values for the spectrum and for the response function, while black lines indicate their "true" values.

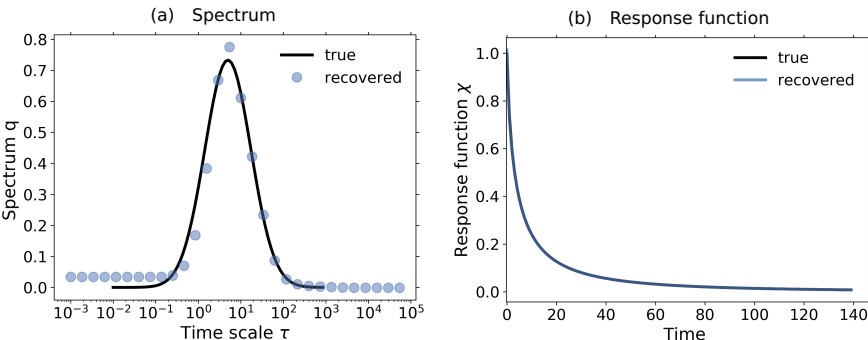

**Figure E9.** Response function $\chi(t)$ and spectrum $q_\lambda$ recovered from toy model data taking the RFI parameters $M = 30$, $\log \tau_{min} = -3$ and $\log \tau_{max} = 5$. Blue dots in (a) and blue line in (b) indicate the recovered values for the spectrum and for the response function, while black lines indicate their "true" values.

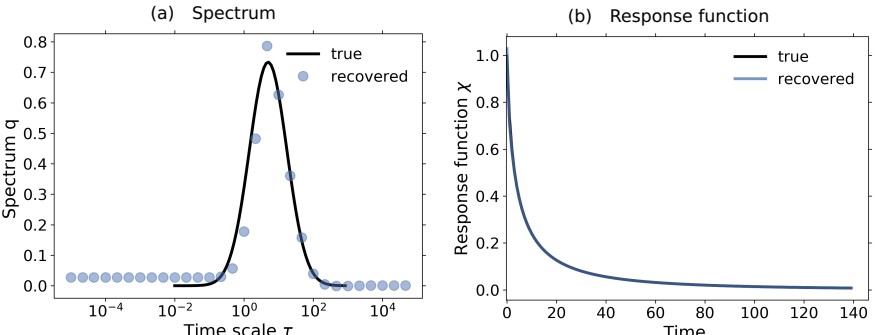

**Figure E10.** Response function $\chi(t)$ and spectrum $q_\lambda$ recovered from toy model data taking the RFI parameters $M = 30$, $\log \tau_{min} = -5$ and $\log \tau_{max} = 5$. Blue dots in (a) and blue line in (b) indicate the recovered values for the spectrum and for the response function, while black lines indicate their "true" values.



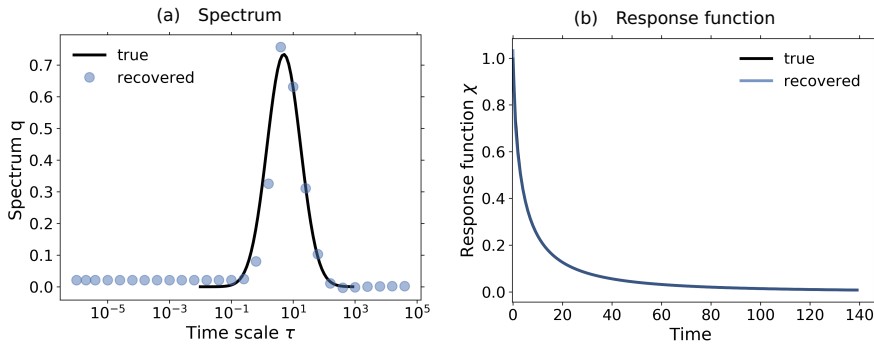

**Figure E11.** Response function $\chi(t)$ and spectrum $\boldsymbol{q}_\lambda$ recovered from toy model data taking the RFI parameters $M = 30$, $\log \tau_{min} = -7$ and $\log \tau_{max} = 5$. Blue dots in (a) and blue line in (b) indicate the recovered values for the spectrum and for the response function, while black lines indicate their "true" values.





*Code and data availability.* The scripts employed to produce the results in this paper as well as information on how to obtain the underlying data can be found at http://hdl.handle.net/21.11116/0000-0008-0F02-6 (Torres Mendonca et al., 2021).

*Author contributions.* The ideas for this study were jointly developed by all authors. GTM conducted the study and wrote the first draft. All authors contributed to the final manuscript.

*Competing interests.* The authors declare that they have no conflict of interest.

*Acknowledgements.* We would like to thank Andreas Chlond for his very helpful suggestions on the manuscript.





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
