# Peer review of "Identification of linear response functions from arbitrary perturbation experiments in the presence of noise Part I. Method development and toy model demonstration"

_Nonlinear Processes in Geophysics, 2021_

## Referee Comment (RC1)

**Identification of linear response functions from arbitrary perturbation experiments in the presence of noise – Part I. Method development and toy model demonstration**

Being a numerical analyst, I am not able to evaluate the application aspect of this manuscript – so I will focus on the methodology and the algorithm.

A large part of the manuscript consists of a review of material that is already well described in the references given by the authors. Since this is not a review paper, I wonder why so much space is devoted to review? I find that due to this lengthy presentation and all the details, it is difficult "to see the forest for all the trees." Specifically, I find it hard to identify precisely what is the new contribution of this work.

The new algorithm is called the "RFI (Response Function Identification) method." This is a very generic name since the goal is, indeed, to solve the deconvolution problem in Eq. (1) – it says nothing about the particular approach taken, and any deconvolution method could go by that name.

The RFI method is summarized in Figure 1, which shows that this is nothing but "plain vanilla" Tikhonov regularization using the discrepancy principle for choosing the regularization parameter. The only novelty seems to be the choice of delta in the discrepancy principle. This could be described much, much shorter.

I honestly do not understand the rationale behind the choice of delta. I can see that delta is the norm of a scaled noise vector, and the scaling depends on an index $i_{max}$ that is "the last index $i$ before the plateau $\sigma_i$ approx 0" [$\sigma_i$ being the singular values of the system matrix]. This means that $i_{max}$ is the number of singular values that are not dominated by rounding errors (and perhaps approximation errors in the discretization). This has nothing to do with the noise in the data, which is the ingredient in the discrepancy principle. That is why I don't understand what is going on here.

Due to the excessive amount of review material, the minimal amount of novelty, and the failure to motivate and explain how delta is computed, I recommend rejection of the manuscript. A much shorter and precise manuscript might be considered for publication.

---

## Author Comment (AC2)

**Response to Anonymous Referee #2 on "Identification of linear response functions from arbitrary perturbation experiments in the presence of noise**

**Part I. Method development and toy model demonstration"**

Guilherme L. Torres Mendonça[1,2], Julia Pongratz[2,3], and Christian H. Reick[2]

[1]International Max Planck Research School on Earth System Modelling, Hamburg, Germany
[2]Max Planck Institute for Meteorology, Hamburg, Germany
[3]Ludwig-Maximillians-Universität München, Munich, Germany

**Correspondence:** Guilherme L. Torres Mendonça (guilherme.mendonca@mpimet.mpg.de)

We would like to thank Anonymous Referee #2 for the positive review of our paper. We largely agree with the referee's suggestions and will address them in the revised manuscript. Below we give a point-by-point reply to the issues raised in the review.

*AR#2*: **In the introduction the authors state that typically methods estimating the response rely on some "prior information". They state that their method does not need prior information. That seems not quite true (they assume monotonicity, Eqn (8), Picard condition etc).**

*Authors*: We appreciate the comment by the referee. Indeed the way these sentences were framed might give the impression that our method does not need prior information. Actually with these (malformed) sentences we tried to point out exactly what the referee requests, namely that additional prior information is needed, particularly in the form of additional data from an unperturbed experiment. But additionally to that, we agree with the referee that we make even more assumptions. We will change the text accordingly.

*AR#2*: **Following up on my previous point, their method requires a few assumptions on the underlying dynamical system under consideration such as (8) and what they call the Picard condition. It would be nice to have these assumptions listed somewhere.**

*Authors*: We agree with the referee and will add such a list to the revised manuscript.

*AR#2*: **It is well known that one can formulate regularization such as ridge regression in a Bayesian framework where the regularization corresponds to a prior. Could the authors comment on what the meaning of this prior is in their context?**

*Authors*: Although we are mostly acquainted with deterministic regularization methods (e.g., Bertero et al., 1995; Engl et al., 1996; Hansen, 2010), it seems to us that from the Bayesian perspective by assuming that the prior distribution for the quantity

of interest $\boldsymbol{q}$ and the distribution for the additive noise $\boldsymbol{\eta}$ are Gaussian (of the type $\mathcal{N}(\boldsymbol{0}, \sigma^2 I)$) and independent, the maximum a posteriori estimator for $\boldsymbol{q}$ gives the same regularized solution that we arrive at (e.g. example 4.26 in Vogel, 2002). But since this is known from the literature and not needed to understand our method, we would prefer not to include such a discussion in the text.

*AR#2*: **Introducing the noise $\eta$ in (5) can be justified by the central limit theorem, I assume, which could be mentioned.**

*Authors*: We do not really understand what the referee is having in mind here. The noise term $\eta$ arises in Eq. (5) simply as the remainder when shifting from the ensemble average to a particular realization. There is no statistical assumption behind this that could be made precise by invoking the central limit theorem. But maybe our formulation in line 134 that the noise term must show up "as a consequence of dropping the ensemble average" is not sufficiently clear. We will think about a better formulation.

*AR#2*: **Although I appreciated that the authors went through some trouble in explaining the details necessary to understand their approach, the manuscript could gain by being more succinct. For example, the sentence right after Eqn (24) just reiterates what has been described before.**

*Authors*: We agree with the reviewer. We will go through the manuscript and see where we can be more succinct.

*AR#2*: **In the introduction the authors give a nice account of the use of linear response theory in the climate sciences. Their exposition, however, might give the false impression that linear response should be expected. Whereas it is now proven that systems driven by noise satisfy linear response theory (Hairer and Majda 2010), the situation for deterministic systems as initiated by Ruelle is far more complicated. Viviane Baladi and co-workers in fact showed that very simple dynamical systems such as the logistic map do not obey linear response. Moreover, examples of dynamical systems in the climate sciences are known that exhibit a rough parameter dependency (Chekroun et al 2014). The question of how to reconcile the fact that generic high-dimensional dynamical systems satisfy linear response theory even when their individual microscopic constituents do not, was addressed by Wormell and Gottwald (2018, 2019). The following references are relevant for this discussion: (...)**

*Authors*: We agree with the reviewer that it is useful to extend the literature review as suggested.

*AR#2*: **There are other recent methods dealing with response theory from a numerical point of view, either detecting it or calculating the response, which the authors may want to include: (...)**

*Authors*: We will refer to these additional numerical methods in the revised manuscript.

**AR#2: Page 15, after Eqn (31): "Since almost every linear system can be diagonalised, we assume" —> "We assume .."**

**AR#2: Page 17, l439, delete "extremely"**

60   *Authors*: The text will be changed accordingly.

With best regards,

Guilherme L. Torres Mendonça, Julia Pongratz and Christian H. Reick

**References**

65    Bertero, M., Boccacci, P., and Maggio, F.: Regularization methods in image restoration: an application to HST images, International Journal of Imaging Systems and Technology, 6, 376–386, 1995.

Engl, H. W., Hanke, M., and Neubauer, A.: Regularization of inverse problems, vol. 375, Springer Science & Business Media, 1996.

Hansen, P. C.: Discrete inverse problems: insight and algorithms, vol. 7, Siam, 2010.

Vogel, C. R.: Computational methods for inverse problems, Siam, 2002.

---

## Author Response (AR1)

**Revision of the manuscript "Identification of linear response functions from arbitrary perturbation experiments in the presence of noise**

**Part I. Method development and toy model demonstration"**

Guilherme L. Torres Mendonça[1,2], Julia Pongratz[2,3], and Christian H. Reick[2]

[1]International Max Planck Research School on Earth System Modelling, Hamburg, Germany
[2]Max Planck Institute for Meteorology, Hamburg, Germany
[3]Ludwig-Maxmillians-Universität München, Munich, Germany

**Correspondence:** Guilherme L. Torres Mendonça (guilherme.mendonca@mpimet.mpg.de)

Dear Editor, dear Reviewers,

We would like to thank you for your valuable suggestions. Below we provide:

1. Comments from the referees and our responses, as posted in the online discussion (in black).

2. Description of the corresponding changes in the manuscript (in blue)*.

3. Manuscript with all revisions highlighted.

In a separate file you find as well the final revised paper.

With best regards,

Guilherme L. Torres Mendonça, Julia Pongratz and Christian H. Reick

Hamburg, June 9, 2021

*When describing the changes in the manuscript, please note that the lines we refer to are those in the manuscript *with the highlighted revisions below* (and not those in the final revised paper attached separately).

**Response to Anonymous Referee #1**

We would like to thank Anonymous Referee #1 for the fast response to our paper, which gives us the opportunity to discuss it more carefully. In view of her/his comments, we believe it is important to first clarify why the paper was framed in the present way and why we are convinced that our method must be introduced at the present level of detail – including a review of some aspects of regularization theory. We thus start with some general remarks clarifying these issues and then proceed to address the referee's comments point by point.

**1   General remarks**

First, although we do believe that with this paper we give a significant contribution to the ill-posed problems literature, we think it should be made clear that this paper was not framed having in mind the ill-posed problems community as the *main* audience. From the point of view of this community, the main novelty of the paper – as correctly pointed out by the referee – is our new approach to estimate the noise in the data to determine the regularization parameter. As we show in more detail in the point-by-point reply, this novelty is indeed emphasized several times throughout the paper.

Nevertheless, the focus of this paper is on the identification of linear response functions – which from the perspective of the ill-posed problems community can be seen as an application of our new approach to estimate the noise level in a regularization procedure to solve a particular ill-posed problem. Accordingly, the paper was framed mainly for a community of applied scientists (particularly in geosciences) interested in studying physical systems by means of linear response functions. From the point of view of these scientists, our new approach to estimate the noise level is only a single step in the RFI algorithm to identify response functions. The real novelty is the RFI method itself, which makes it possible for the first time to identify these functions taking noisy data from arbitrary perturbation experiments. Therefore, although we do emphasize several times that the main novelty of the method is our noise estimation approach, the main focus of the paper is on the resulting RFI method and how with this method one can identify linear response functions from perturbation experiment data. We believe that the fact that we focus on the resulting RFI method instead of only on the noise estimation procedure might have made it difficult for the referee to "see the forest for all the trees", since she/he evaluated only the methodology and the algorithm, and therefore could see the novelty only from the point of view of the contribution to the ill-posed problems literature.

The fact that this paper was framed mostly for a community interested in the response function approach also explains why we introduce our RFI method starting from a review of details that are well known to the ill-posed problems community. As discussed in the introduction of the paper, current methods in geosciences to identify linear response functions from data do not explicitly account for the ill-posedness of the identification problem. This indicates that the ill-posedness of this problem is not completely clear to at least a large part of the geosciences community interested in the response function approach. Hence, this ill-posedness is already an issue that must be explained to that audience. But most importantly, if one is not aware of the whole ill-posedness issue, one is probably not aware either of details of regularization theory, which is a means to treat that

issue. Since for understanding our RFI method one must understand particular aspects of the large corpus of knowledge on regularization (like the discrete Picard condition that is necessary for understanding the noise estimation explained in section 3.4, and "Hansen's observation", needed to understand the additional noise level adjustment in section 3.5), to get our paper understood by our main audience we therefore find it essential to clearly explain those aspects. Further, because our method relies so much on those particular details, we believe that such review is beneficial even for people already working on ill-posed problems that may get interested in our method.

With these general remarks we now proceed to respond the reviewer's comments point by point.

**2 Point-by-point reply**

*AR#1*: **Being a numerical analyst, I am not able to evaluate the application aspect of this manuscript – so I will focus on the methodology and the algorithm. A large part of the manuscript consists of a review of material that is already well described in the references given by the authors. Since this is not a review paper, I wonder why so much space is devoted to review?**

*Authors*: To answer this comment we refer to our general remarks above. In summary, as pointed out in the introduction of the paper, the main motivation for the design of this method is its practical application to study physical systems. Since scientists interested in applications are not necessarily aware of the nuances of regularization theory, we are convinced that an introduction to the aspects of the theory that are relevant for understanding our method is essential to get those scientists into the boat. In addition, because our method is so deeply rooted in certain details of regularization theory, we believe that explicitly stating those details is beneficial even for initiated readers. Still, we agree that it would make sense to make more clear what parts are review, and where the novelty begins.

*Changes*: We revised the manuscript to make clearer where we are reviewing known aspects from regularization theory, and where the novelty of our method begins. The review is now presented in the separate section 3.3 (see highlighted revisions in the manuscript below: L266, L268–273, L321, L322, L332–334), and our new approach to estimate the noise level is presented in sections 3.4 and 3.5 (minor highlighted revisions: L335, L336, L340). We also emphasize the main novelty of the study in the introduction of section 3 where we derive the method (see highlighted revisions: L175, L176, L186, L188–193).

*AR#1*: **I find that due to this lengthy presentation and all the details, it is difficult "to see the forest for all the trees." Specifically, I find it hard to identify precisely what is the new contribution of this work.**

*Authors*: We really regret that the referee finds it difficult "to see the forest for all the trees". As stated in the general remarks above, we believe that in part this difficulty is explained by noting that our algorithmic improvements cannot be appreciated without considering also the application aspect of our study, namely the identification of linear response functions, which is our main focus and presents the most relevant novelty to our targeted audience – a method to identify linear response functions taking noisy data from arbitrary perturbation experiments. Still, obviously the main novel idea of this method – which *was* evaluated by the referee – to combine information from a SVD analysis of the perturbation experiment data and from an additional unperturbed data stream to more objectively estimate the noise level and thereby the regularization parameter could not be sufficiently conveyed. Nevertheless, we are surprised by this comment, because we repeatedly emphasize this novel idea, e.g. in the abstract (L5-7 and L11-13), in the introduction (p. 4, L103-105), in section 2 (p. 5, L144-146), when introducing section 3 (p. 6, L152-155, L158-162, L165-166), in section 3.4 (where the idea is in detail explained), in section 4 (where we numerically demonstrate how the noise estimation works), and in section 6 (p. 31-32, L744-748; p. 33, L759-778; p. 34, L779-790).

*Changes*: No action taken.

*AR#1*: **The new algorithm is called the "RFI (Response Function Identification) method." This is a very generic name since the goal is, indeed, to solve the deconvolution problem in Eq. (1) – it says nothing about the particular approach taken, and any deconvolution method could go by that name.**

*Authors*: Only for response problems one can expect to have an additional unperturbed data stream from an independent data source to determine the regularization parameter, which is not the case for general deconvolution problems. In our view, for this reason the choice of the name makes sense instead of simply considering the RFI method as a general deconvolution method. Nevertheless, we are of course open to suggestions from the referee for a more appropriate name.

*Changes*: No action taken.

*AR#1*: **The RFI method is summarized in Figure 1, which shows that this is nothing but "plain vanilla" Tikhonov regularization using the discrepancy principle for choosing the regularization parameter. The only novelty seems to be the choice of delta in the discrepancy principle. This could be described much, much shorter.**

*Authors*: The referee is correct in pointing out that the novelty of the method lies in the estimation of the noise level $\delta$ for determining the regularization parameter. We refer to our general remarks and to our first response as to why we believe it is important to introduce also aspects of the method that are not new from the point of view of the ill-posed problems community.

*Changes*: No action taken.

*AR#1*: **I honestly do not understand the rationale behind the choice of delta. I can see that delta is the norm of a scaled noise vector, and the scaling depends on an index i_max that is "the last index i before the plateau sigma_i approx 0" [sigma_i being the singular values of the system matrix]. This means that i_max is the number of singular values that are not dominated by rounding errors (and perhaps approximation errors in the discretization). This has nothing to do with the noise in the data, which is the ingredient in the discrepancy principle. That is why I don't understand what is going on here.**

*Authors*: We fully agree with the reviewer's interpretation of $i_{max}$ that this index gives the number of singular values that are not dominated by rounding and maybe discretization errors. Nevertheless, as we explain in the paper (section 3.4) and emphasize below, by determining $i_{max}$ one can obtain important information about the noise in the data, namely the range of index values for which the SVD components of the data contain for sure only noise.

120  The rationale for our approach to estimate $\delta$ is in detail explained in sections 3.4 and 3.5. Specifically, the explanation for this estimation starts in p. 11, L312. How the approach works numerically is also carefully discussed in the examples of section 4, e.g. in Fig. 5, where we demonstrate how step 3 of the algorithm works (i.e. the scaling of the noise taken from the control experiment, introduced in section 3.4); in Fig. 6, where we show how step 6 works (i.e. the additional noise level adjustment in the presence of a monotonicity constraint, introduced in section 3.5); and in Fig. 8, where we discuss how our noise estimation

125  procedure behaves in the presence of nonlinearities.

  But to make our approach even clearer, in the following we visually explain its basic steps, once more using a numerical example. The data for the example were taken from the same toy model experiment used for the analysis of Fig. 5 in the paper. Our step-by-step explanation is shown in Figs. 1–6 below, in a sequence of Picard plots with a text description of each of the steps 1–3 of our RFI algorithm – the basic steps performed to estimate $\delta$. For the particular case where the response function

130  is monotonic, this estimate may be even further improved by the additional step 6 (described in section 3.5 of the paper).

  More technically, the approach is the following:

- In the first step, we take a first-order noise estimate $\boldsymbol{\eta}_{ctrl}$ obtained from the additional unperturbed data stream $\boldsymbol{\Delta Y}_{ctrl}$ – the data taken from the *control experiment* (L328–330 of the paper; Fig. 2 in the present reply).

- In the second step, we define $i_{max}$ as the last index before the plateau $\sigma_i \approx 0$ (L337 of the paper; Fig. 3 in the present
135  reply).

- We then note the following: Since by definition for $i > i_{max}$ the singular values are $\sigma_i \approx 0$, by the Picard condition also the projection coefficients of the "clean" data $\boldsymbol{u}_i \bullet \mathbf{A}\boldsymbol{q}$ must have dropped to zero. As a result – as shown by Eq. (25) of the paper –, for $i > i_{max}$ the projection coefficients of the data $\boldsymbol{u}_i \bullet \boldsymbol{\Delta Y}$ are completely dominated by the noise contribution $\boldsymbol{u}_i \bullet \boldsymbol{\eta}$. Therefore, for $i > i_{max}$ the data components $\boldsymbol{u}_i \bullet \boldsymbol{\Delta Y}$ can be taken as an estimate of the respective
140  noise components $\boldsymbol{u}_i \bullet \boldsymbol{\eta}$ (L312–327 of the paper; Fig. 4 in the present reply).

- In the third step, we (i) collect the SVD coefficients $\boldsymbol{u}_i \bullet \boldsymbol{\Delta Y}$ and $\boldsymbol{u}_i \bullet \boldsymbol{\eta}_{ctrl}$ for $i > i_{max}$ in two vectors, and compute their norms $z$ and $z_{ctrl}$ (L338–341 in the paper; Fig. 5 here); and then (ii) scale by $z/z_{ctrl}$ the noise from the control experiment $\boldsymbol{\eta}_{ctrl}$ to obtain $\boldsymbol{\eta}'$ (L341–343 in the paper; Fig. 6 here).

- In this way, the magnitude of the SVD components for $i > i_{max}$ of $\boldsymbol{\eta}'$ matches that of $\boldsymbol{\Delta Y}$, and, because of Eq. (25),
145  also that of $\boldsymbol{\eta}$. If the spectral distribution of $\boldsymbol{\eta}_{ctrl}$ is similar to that of $\boldsymbol{\eta}$ (spectral similarity assumption), then $\boldsymbol{\eta}'$ can be seen as an estimate for $\boldsymbol{\eta}$ (L344–L348 of the paper; also Fig. 6 here).

- With the resulting noise estimate $\boldsymbol{\eta}'$ we finally set the noise level $\delta := ||\boldsymbol{\eta}'||$, which is then used in the discrepancy method (L349-354 of the paper).

- If monotonicity of the response function should be accounted for, the resulting estimate of $\delta$ is further adjusted in an
150  iterative way until the constraint is enforced (section 3.5 of the paper).

*Changes*: This step-by-step answer to the reviewer is a kind of extract from our paper answering particularly his irritation on the meaning of $i_{max}$. Therefore it is too specific to be taken over into the paper. Moreover, we do not see how we could be even more explicit in the paper to fully prevent the irritation experienced by the reviewer. Therefore we took no action on this point.

[Figure]

**Figure 1.** Picard plot for exemplary data taken from a simulation with our toy model (described in section 4.1 of the paper). The data vector $\mathbf{\Delta Y}$ from the perturbation experiment are in magenta. In green are the singular values of the matrix $\mathbf{A}$, and in blue the noise from the control experiment $\boldsymbol{\eta}_{ctrl}$.

[Figure]

**Figure 2.** First step of the RFI algorithm: The noise from the control experiment $\eta_{ctrl}$ is taken from control experiment data $\mathbf{\Delta Y}_{ctrl}$.

[Figure]

**2.** $i_{max}$ is defined as the last index before the plateau $\sigma_i \approx 0$ . This index distinguishes high-frequency from low-frequency components.

**Figure 3.** Second step of the RFI algorithm: The index $i_{max}$ is determined as the last index $i$ before the plateau $\sigma_i \approx 0$.

[Figure]

For $i > i_{max}$, $\sigma_i \approx 0$ and by the discrete Picard condition also the components of the "clean data" $|u_i \bullet Aq| \approx 0$. Hence, for $i > i_{max}$ we can be sure that the perturbation experiment data are dominated by noise, i.e.

$$u_i \bullet \Delta Y \approx u_i \bullet \eta \quad \text{(Eq. (25))}$$

Legend:
- $\sigma_i$
- $|u_i \cdot \Delta Y|$
- $|u_i \cdot \eta|$
- $|u_i \cdot \eta_{ctrl}|$

true noise in the perturbation experiment data

$i_{max} = 21$

plateau $\sigma_i \approx 0$

$i > i_{max}$

**Figure 4.** Explanation for the relevance of $i_{max}$ to determine the noise in the data: For $i > i_{max}$, the data are dominated by noise. Therefore, for $i > i_{max}$, the data components $u_i \bullet \Delta Y$ can be understood an estimate of the noise components $u_i \bullet \eta$.

**3.** Define $z$ as the norm of the high-frequency components of the perturbation experiment data, which by Eq. (25) are also the high-frequency components of the noise in the data. Define $z_{ctrl}$ as the norm of the high-frequency noise components from the control experiment.

[Figure]

$$z := ||[\mathbf{u}_{i_{max}+1} \bullet \mathbf{\Delta Y}, ..., \mathbf{u}_{30} \bullet \mathbf{\Delta Y}]||$$

$$z_{ctrl} := ||[\mathbf{u}_{i_{max}+1} \bullet \eta_{ctrl}, ..., \mathbf{u}_{30} \bullet \eta_{ctrl}]||$$

- $\sigma_i$
- $|u_i \cdot \Delta Y|$
- $|u_i \cdot \eta_{ctrl}|$

**Figure 5.** Third step of the RFI algorithm: Define $z$ as the norm of the high-frequency components of the data $\mathbf{\Delta Y}$, which by Eq. (25) of our paper are also the high-frequency components of the noise $\eta$. In addition, define $z_{ctrl}$ as the norm of the high-frequency components of the noise from the control experiment $\eta_{ctrl}$.

**3.** Using $z$ and $z_{ctrl}$, scale the noise from the control experiment $\eta_{ctrl}$ so that its high-frequency level matches that of the noise in the perturbation experiment data $\eta$.

$$\eta' := \frac{z}{z_{ctrl}} \eta_{ctrl}$$

[Figure]

If the noise from the control experiment $\eta_{ctrl}$ has a spectral distribution (the "shape" of the spectrum) similar to that of the noise in the perturbation experiment data $\eta$ (i.e., if the *spectral similarity assumption* holds), then the resulting scaled noise vector $\eta'$ is a reasonable estimate for $\eta$. Therefore we set the noise level $\delta := ||\eta'||$, which is then used in the discrepancy method.

**Figure 6.** Third step of the RFI algorithm (continuing): scale by $z/z_{ctrl}$ the noise from the control experiment $\eta_{ctrl}$ to obtain $\eta'$. If the spectral distribution of $\eta_{ctrl}$ is similar to that of $\eta$ (spectral similarity assumption), then $\eta'$ can be seen as an estimate for $\eta$.

**AR#1**: *Due to the excessive amount of review material, the minimal amount of novelty, and the failure to motivate and explain how delta is computed, I recommend rejection of the manuscript. A much shorter and precise manuscript might be considered for publication.*

*Authors*: We still think that our study represents an important methodological advancement not only for the geosciences community interested in the response function approach – whose viewpoint is the main focus of the manuscript –, but also for scientists interested in ill-posed problems. As demonstrated by the large number of studies employing response functions in geosciences, especially in recent years – see introduction, in particular L35-41 –, these functions represent a powerful tool of increasing importance for the geosciences community. But as also discussed in the introduction, currently no method is available in the field to identify response functions taking data from any arbitrary type of perturbation experiment. Further, even in the case where a tailored perturbation experiment for identifying these functions is available, noise in the data usually hinders a reliable identification. This often makes it necessary to perform many experiments to obtain a better signal-to-noise ratio. These two difficulties (the need for a special perturbation experiment and for sufficiently "clean" data) thus severely restrict the applicability of the response function approach. Hence, by presenting a method to identify response functions from arbitrary perturbation experiments in the presence of noise, we believe that this paper gives a relevant contribution to the field that allows for a much wider applicability of the response function approach.

From the point of view of scientists interested in ill-posed problems, we are convinced that our method represents also an important advancement. The reason is that we present for the problem of response functions identification an approach to estimate the noise in the data on more objective grounds. As is known from the literature – and also discussed in our paper (see section 6, starting from L759) –, to obtain a regularized solution that converges to the "true" solution of the problem for decreasing noise level, regularization methods need to account for the noise level (Bakushinskii, 1984). But in practice this noise level is rarely known with accuracy: In fact, several studies investigate the typical situation where one has only a guess of the noise level at hand (e.g., Raus, 1992; Hämarik and Raus, 2006; Hämarik et al., 2011, 2012). With our approach, this noise level can in principle be more accurately estimated by using information from a SVD analysis of the data and from an additional unperturbed data stream: This is numerically demonstrated not only by the results of the application of our method to toy model simulations shown in the present paper, but also by the results of its application to a real problem in Part II. Further, although our method was designed for the identification of linear response functions, as discussed in section 6 (starting from L779) it may in principle find application in solving also other types of linear ill-posed problems.

But to make these advancements more obvious, it may indeed be useful to convey better to the reader what parts of our paper are reviewing standard knowledge in numerical analysis, and what is new.

*Changes*: As described more specifically above, we revised the manuscript to show more clearly where we are only reviewing existent knowledge on regularization, and where the novelty of our method, namely our new approach to estimate the noise level, starts. The review is now presented in the separate section 3.3, and our new approach is introduced in sections 3.4 and 3.5. We also emphasize the main novelty of the study in the introduction of section 3 where we derive the method. For more details please see the revised manuscript with the highlighted changes.

**Response to Anonymous Referee #2**

We would like to thank Anonymous Referee #2 for the positive review of our paper. We largely agree with the referee's suggestions and will address them in the revised manuscript. Below we give a point-by-point reply to the issues raised in the review.

*AR#2*: **In the introduction the authors state that typically methods estimating the response rely on some "prior information". They state that their method does not need prior information. That seems not quite true (they assume monotonicity, Eqn (8), Picard condition etc).**

*Authors*: We appreciate the comment by the referee. Indeed the way these sentences were framed might give the impression that our method does not need prior information. Actually with these (malformed) sentences we tried to point out exactly what the referee requests, namely that additional prior information is needed, particularly in the form of additional data from an unperturbed experiment. But additionally to that, we agree with the referee that we make even more assumptions. We will change the text accordingly.

*Changes*: We reformulated the referred sentence to state more explicitly that our method does need prior information (see highlighted revisions in the manuscript below: L117–120).

*AR#2*: **Following up on my previous point, their method requires a few assumptions on the underlying dynamical system under consideration such as (8) and what they call the Picard condition. It would be nice to have these assumptions listed somewhere.**

*Authors*: We agree with the referee and will add such a list to the revised manuscript.

*Changes*: The list of underlying assumptions has been added to the manuscript (see L422–424 and Table 1 in the text with highlighted revisions).

*AR#2*: **It is well known that one can formulate regularization such as ridge regression in a Bayesian framework where the regularization corresponds to a prior. Could the authors comment on what the meaning of this prior is in their context?**

*Authors*: Although we are mostly acquainted with deterministic regularization methods (e.g., Bertero et al., 1995; Engl et al., 1996; Hansen, 2010), it seems to us that from the Bayesian perspective by assuming that the prior distribution for the quantity

of interest $q$ and the distribution for the additive noise $\eta$ are Gaussian (of the type $\mathcal{N}(\mathbf{0}, \sigma^2 I)$) and independent, the maximum a posteriori estimator for $q$ gives the same regularized solution that we arrive at (e.g. example 4.26 in Vogel, 2002). But since this is known from the literature and not needed to understand our method, we would prefer not to include such a discussion in the text.

*Changes*: No action taken.

225

*AR#2*: **Introducing the noise $\eta$ in (5) can be justified by the central limit theorem, I assume, which could be mentioned.**

*Authors*: We do not really understand what the referee is having in mind here. The noise term $\eta$ arises in Eq. (5) simply as the remainder when shifting from the ensemble average to a particular realization. There is no statistical assumption behind

230   this that could be made precise by invoking the central limit theorem. But maybe our formulation in line 134 that the noise term must show up "as a consequence of dropping the ensemble average" is not sufficiently clear. We will think about a better formulation.

*Changes*: We added a more explicit explanation for how we define the noise term (L155–157 in the revised text below).

235   *AR#2*: **Although I appreciated that the authors went through some trouble in explaining the details necessary to understand their approach, the manuscript could gain by being more succinct. For example, the sentence right after Eqn (24) just reiterates what has been described before.**

*Authors*: We agree with the reviewer. We will go through the manuscript and see where we can be more succinct.

*Changes*: We deleted the referred sentence (L357–359), and also additional considerations about the ill-posedness of the iden-

240   tification problem at the beginning of section 4.4 (L490–496).

*AR#2*: **In the introduction the authors give a nice account of the use of linear response theory in the climate sciences. Their exposition, however, might give the false impression that linear response should be expected. Whereas it is now proven that systems driven by noise satisfy linear response theory (Hairer and Majda 2010), the situation for deter-**

245   **ministic systems as initiated by Ruelle is far more complicated. Viviane Baladi and co-workers in fact showed that very simple dynamical systems such as the logistic map do not obey linear response. Moreover, examples of dynamical systems in the climate sciences are known that exhibit a rough parameter dependency (Chekroun et al 2014). The question of how to reconcile the fact that generic high-dimensional dynamical systems satisfy linear response theory even when their individual microscopic constituents do not, was addressed by Wormell and Gottwald (2018, 2019). The following**

250   **references are relevant for this discussion: (...)**

*Authors*: We agree with the reviewer that it is useful to extend the literature review as suggested.

*Changes*: We take this remark as a suggestion to improve the literature overview. To put the suggested papers into better context, we now cite also some other papers and books. Note that instead of citing all the suggested papers by Baladi separately, we cite instead her monograph on these subjects, that serves also as a rather comprehensive guide to her work on linear response.

255   The corresponding revisions can be found in L26–62.

*AR#2*: **There are other recent methods dealing with response theory from a numerical point of view, either detecting it or calculating the response, which the authors may want to include: (...)**

*Authors*: We will refer to these additional numerical methods in the revised manuscript.

260 *Changes*: Done (L89–92).

*AR#2*: **Page 15, after Eqn (31): "Since almost every linear system can be diagonalised, we assume" —> "We assume .."**
*AR#2*: **Page 17, l439, delete "extremely"**

*Authors*: The text will be changed accordingly.

265 *Changes*: Done (L439, L477).

**Response to the comment by Valerio Lucarini**

270

*Valerio Lucarini*: **I have greatly appreciated your paper and the idea behind it. The application developed in Part II is also very interesting. I have a comment I would recommend you to consider. When you introduce Eqs. 7 and 8 you are making the (very important) assumption that all the eigenvalues of the unperturbed transfer operator are real, whereas they are in general complex (with each complex number accompanied by its conjugate). Complex conjugate pairs allow**

275 **for the presence of oscillating terms in the Green function. You might see a longer explanation of this in Tantet et al. 2020 and Lucarini 2018.. In other terms, you are allowing for purely relaxation behvaiour in your system. This might well work out fine for some applications, but not on other ones (where more complex feedbacks are present).**

*Authors*: We appreciate your comments. We fully agree with your remark that our Eq. (8) assumes real relaxation times $\tau$ and thereby rules out oscillatory behaviour. We point out this limitation in the discussions section, L822-824. Also referee #2

280 asked us to make the assumptions underlying our approach more transparent. Following that suggestion we will mention in the introduction and also when introducing Eq. (8) that the respective eigenvalues are assumed to be real. This indeed introduces an important limitation, namely to the class of overdamped systems to which presumably the global carbon cycle belongs, which is our main application in Part II of this study. We emphasize however that, as discussed in L804-824, Eq. (8) is not essential for our RFI method, so that by dropping this assumption (thus recovering the response function pointwise) one could

285 in principle still apply the method to more general systems.

Nevertheless, as also discussed in L804-824, assuming Eq. (8) has some advantages over recovering the response function pointwise. Therefore it would be nice to be able apply our RFI method by extending this assumption to include complex eigenvalues. But at the moment it is not clear to us how this could be done.

*Changes*: We mention now in the introduction (L120–123 in the revisions below), when presenting Eq. (8) (L222, L223) and

290 also in the list of assumptions in Table 1 that we assume that the response is dominated by non-oscillatory relaxation modes. We also reformulated our remark on this limitation in the discussions section (L861–863).

[revised manuscript text omitted]

---

## Author Response (AR2)

**Revision of the manuscript "Identification of linear response functions from arbitrary perturbation experiments in the presence of noise**

**Part I. Method development and toy model demonstration"**

Guilherme L. Torres Mendonça[1,2], Julia Pongratz[2,3], and Christian H. Reick[2]

[1]International Max Planck Research School on Earth System Modelling, Hamburg, Germany
[2]Max Planck Institute for Meteorology, Hamburg, Germany
[3]Ludwig-Maxmillians-Universität München, Munich, Germany

**Correspondence:** Guilherme L. Torres Mendonça (guilherme.mendonca@mpimet.mpg.de)

Dear Editor,

We thank you and the reviewers for reconsidering our paper. According to your comment, there is a single issue remaining: For resubmission you asked us to account for the comment by reviewer #3, to check our paper for unnecessary verbosity to make it more concise. Following this suggestion, we once more went through our paper. But because we already tried to shorten the paper during the first round of reviews, we do not see much potential for a substantial reduction. Nevertheless, we still identified a few sentences and paragraphs that also in our view can be be shortened or deleted.

Below we document these changes by

1. Reproducing the comment by reviewer #3 and adding a short response to it (in black).

2. Describing the corresponding changes in the manuscript (in blue).

3. Appending the revised manuscript with all changes highlighted.

In a separate file you find as well the final revised paper.

With best regards,

Guilherme L. Torres Mendonça, Julia Pongratz and Christian H. Reick

Hamburg, August 2, 2021

**Response to Anonymous Referee #3**

25    We would like to thank Anonymous Referee #3 for the overall positive review of our paper. Below we give a detailed reply to the comments.

*AR#3*: **The authors present a data driven means of determining linear response functions from judiciously designed perturbation and control experiments, with application to a toy model. The research is sound, and I have nothing of**

30    **substance further to add that has not already been addressed by the other reviewers. The manuscript is sufficiently well written, but often unnecessarily verbose. For this paper to be well cited in the future, its readability is key. I feel that taking on board some of reviewer 1's comments in particular more seriously would have further improved the manuscript in this regard. A manuscript is not complete when there is nothing more to add, rather it is complete when there is nothing more to remove. I suggest the authors review their manuscript again with this adage in mind. Despite**

35    **the above issues, I recommend the paper for publication. I'd suggest, though it is in the authors' best interest to revise their manuscript to be as concise as possible.**

*Authors*: We agree that we should strive for maximum conciseness. Therefore, as requested, we went back to the comments by reviewer #1 who asked us in particular to shorten our exposition of regularization theory. But as we had already expressed in our response to referee #1, we still believe that a review of certain details of regularization theory is crucial for the understand-

40    ing of our method, even if that makes the paper longer. Still, we agree that it makes sense to revise the paper once more to see where we can make it more concise.

*Changes*: We have reviewed the manuscript and excluded/changed certain excerpts to try to make the content less verbose (see manuscript with highlighted revisions: L31, L173–174, L182, L260–261, L268, L296–298, L308–311, L574–L577, L617–L625).

[revised manuscript text omitted]